UPDATE ARTICLE

# Maternal TGF-β ligand Panda breaks the radial symmetry of the sea urchin embryo by antagonizing the Nodal type II receptor ACVRII

**Praveen Kumar Viswanathan, Aline Chessel, Maria Dolores Molina¤, Emmanuel Haillot, Thierry Lepage** *

Université Côte d'Azur, CNRS, Inserm, iBV, Nice, France

¤ Current address: Department of Genetics, Microbiology and Statistics, Faculty of Biology, University of Barcelona, Barcelona, Catalonia, Spain
* thierry.lepage@univ-cotedazur.fr

## Abstract

In the highly regulative embryo of the sea urchin *Paracentrotus lividus*, establishment of the dorsal-ventral (D/V) axis critically depends on the zygotic expression of the TGF-β *nodal* in the ventral ectoderm. *nodal* expression is first induced ubiquitously in the 32-cell embryo and becomes progressively restricted to the presumptive ventral ectoderm by the early blastula stage. This early spatial restriction of *nodal* expression is independent of Lefty, and instead relies on the activity of Panda, a maternally expressed TGF-β ligand related to Lefty and Inhibins, which is required maternally for D/V axis specification. However, the mechanism by which Panda restricts the early *nodal* expression has remained enigmatic and it is not known if Panda works like a BMP ligand by opposing Nodal and antagonizing Smad2/3 signaling, or if it works like Lefty by sequestering an essential component of the Nodal signaling pathway. In this study, we report that Panda functions as an antagonist of the TGF-β type II receptor ACVRII (Activin receptor type II), which is the only type II receptor for Nodal signaling in the sea urchin and is also a type II receptor for BMP ligands. Inhibiting translation of *acvrII* mRNA disrupted D/V patterning across all 3 germ layers and caused *acvrII* morphants to develop with a typical Nodal loss-of-function phenotype. In contrast, embryos overexpressing *acvrII* displayed strong ectopic Smad1/5/8 signaling at blastula stages and developed as dorsalized larvae, a phenotype very similar to that caused by over activation of BMP signaling. Remarkably, embryos co-injected with *acvrII* mRNA and *panda* mRNA did not show ectopic Smad1/5/8 signaling and developed with a largely normal dorsal-ventral polarity. Furthermore, using an axis induction assay, we found that Panda blocks the ability of ACVRII to orient the D/V axis when overexpressed locally. Using co-immunoprecipitation, we showed that Panda physically interacts with ACVRII, as well as with the Nodal co-receptor Cripto, and with TBR3 (Betaglycan), which is a non-signaling receptor for Inhibins in mammals. At the molecular level, we have traced back the antagonistic activity of Panda to the presence of a single proline residue, conserved with all the Lefty factors, in the ACVRII binding motif of Panda, instead of a serine as in most of TGF-β ligands. Conversion of this proline to a serine converted Panda from an antagonist that opposed Nodal signaling and promoted dorsalization to an agonist that promoted Nodal signaling and triggered

The Editors encourage authors to publish research updates to this article type. Please follow the link in the citation below to view any related articles.

**Data Availability Statement:** All relevant data are within the paper and its Supporting Information files.

**Funding:** This research was funded by the Foundation for Medical Research (FRM), (Grant DEQ20180339195 to TL), by the Foundation for Cancer research (ARC) (Grant: ARCPJA32020060002217) to TL, by the Centre National de la Recherche Scientifique (CNRS) to TL, and by a grant from the Agence Nationale de la Recherche (ANR) to TL (ANR-14-CE11-0006-01). PV was supported by a 3-year PhD fellowship from the Foundation for Medical Research (FRM), (Grant DEQ20180339195 to TL) and by a 4th year PhD fellowship from the Ligue nationale contre le cancer. EH was supported by a grant from the Ministère de la Recherche et de l'Enseignement Supérieur and by a 4th year of PhD fellowship from the ARC. MDM was supported by an European Molecular Biology Organization (EMBO) long-term fellowship (grant 1234-2011) and by an ARC postdoctoral fellowship (grant 2011-1204261). The funders had no role in study design, data collection and analysis, decision to publish, or preparation of the manuscript.

**Competing interests:** The authors have declared that no competing interests exist.

**Abbreviations:** aLRT, approximate likelihood ratio test; hpf, hours post fertilization; PBS, phosphate buffer saline; PMC, primary mesenchymal cell.

ventralization when overexpressed. Finally, using phylogenomics, we analyzed the emergence of the agonist and antagonist form of Panda in the course of evolution. Our data are consistent with the idea that the presence of a serine at that position, like in most TGF-β, was the ancestral condition and that the initial function of Panda was possibly in promoting and not in antagonizing Nodal signaling. These results highlight the existence of key functional and structural elements conserved between Panda and Lefty, allow to draw an intriguing parallel between sea urchin Panda and mammalian Inhibin α and raise the unexpected possibility that the original function of Panda may have been in activation of the Nodal pathway rather than in its inhibition.

## Introduction

The dorsal-ventral (D/V) axis, which is orthogonal to the A/P axis, is one of the main axes of polarity of the embryo. It identifies the back and the belly of the embryo and allows subsequent establishment of the left and right sides. Specification of the D/V axis is a key event during establishment of the body plan of bilaterians. In some species such as *Drosophila*, this process relies on the presence of maternal determinants deposited in discrete regions of the egg. For example, in *Drosophila*, specification of the D/V axis is initiated by a gradient of the maternal factor Dorsal which is itself established by the local accumulation of the Spaetzle morphogen in the perivitelline space [1].

In other animals, such as in mammals, there is very little evidence for the presence of maternal determinants of the D/V axis in the oocyte suggesting that, in these organisms, the D/V axis is specified by cell interactions. Similarly, in the highly regulative sea urchin embryo, the high plasticity of the early blastomeres, revealed by the classical experiments from Driesch and Hörstadius, seemed to exclude the possibility that maternal determinants regulate the D/V axis specification in the blastomeres that inherited them [2]. In support of this idea, in his classical blastomere dissociation, Driesch showed that each blastomere dissociated at the 2- or 4-cell stage has the potential to re-establish a D/V axis, suggesting that the D/V axis is not strongly specified by localized maternal determinants. However, egg-bisection experiments performed by Hörstadius showed that differences between presumptive dorsal and ventral territories could be traced back to the egg, suggesting a preexisting bilateral symmetry in the egg [3] and leaving open the question of the contribution of maternal factors to the determination of the D/V axis.

### The early spatial restriction of *nodal* expression as a key step in establishment of the D/V axis

Consistent with the idea that the D/V axis of the sea urchin embryo is established after fertilization, specification of the D/V axis critically relies on the zygotic expression of the gene encoding the TGF-β ligand Nodal in the presumptive ventral ectoderm [4]. The ventral ectoderm, which expresses *nodal*, acts as an organizing center by inducing the expression of ventral genes and by triggering expression of *bmp2/4*. Once translated, BMP2/4 protein is translocated to the dorsal side and acts as a long-range morphogen to pattern the dorsal territory [5,6]. As a result, embryos that lack Nodal function fail to establish any D/V polarity, are highly pigmented, appear strongly radialized, and lose expression of both ventral and dorsal genes. However, injection of *nodal* mRNA into one blastomere at the eight-cell stage is capable of completely rescuing D/V polarity in these embryos indicating that localized *nodal* expression

is sufficient to induce a D/V axis. On the contrary, overexpression of *nodal* results in albino embryos that are ventralized. Therefore, the spatially restricted expression of *nodal* is the key event that initiates the ventral and dorsal gene regulatory networks that pattern the embryo along the D/V axis. However, although many zygotic genes that act downstream of Nodal to pattern the D/V axis have been identified, the maternal factors and the mechanisms that regulate the initial spatial expression of *nodal* are not well understood [7–12].

## The spatial restriction of *nodal* expression involves a step of symmetry breaking by maternal factors followed by a reaction–diffusion mechanism between Nodal and Lefty

*nodal* is the first zygotic gene that is asymmetrically expressed along the D/V axis. *nodal* expression starts around the 32-cell stage but *nodal* expression is initially activated in most cells of the embryos and, in most embryos, in a broad D/V gradient [4,9]. By the early blastula stage, however, *nodal* expression is sharply restricted to the presumptive ventral ectoderm. The spatial restriction of the *nodal* expression domain in the sea urchin embryo critically relies on the ability of Nodal to promote its own expression and on that of the long-range and highly diffusible Nodal antagonist and Nodal target gene Lefty to restrict it [9]. These regulatory mechanisms serve as the basis for a reaction–diffusion mechanism that is thought to amplify an initial weak asymmetry generated by maternal factors into a robust spatially restricted expression of *nodal* [13]. The asymmetry of *nodal* expression is observed slightly before the onset of *lefty* expression, which is induced by Nodal signaling [4,14]. This suggests that although Lefty is needed to maintain the asymmetry of *nodal* expression, it is probably not responsible for initiating the early spatial restriction of *nodal*.

## The activities of the maternal BMP type I receptors Alk1/2 and Alk3/6 are also required for the spatial restriction of *nodal* expression

In many vertebrate species, patterning of the embryo along the D/V axis depends on a mutual antagonism between 2 signaling centers: the dorsal signaling center that expresses Nodal and BMP antagonists, and the ventral signaling center that expresses BMP ligands. Indeed, the early spatial restriction of *nodal* in the sea urchin embryo also relies on the activity of the BMP type I receptors Alk1/2 and Alk3/6 [8]. The double inhibition of Alk1/2 or Alk3/6 by morpholino injection caused a massive ectopic expression of *nodal*. The finding that knocking down the 2 BMP type I receptors caused a phenotype that was visible even before the onset of *bmp2/4* expression, the main player acting in D/V axis formation, strongly suggested that another TGF-β ligand, possibly maternally expressed, was cooperating with BMP2/4 and acting possibly through the BMP type I receptors Alk1/2 and Alk3/6 to restrict *nodal* expression. This idea led us to search for TGF-β ligands expressed maternally that would be required for D/V axis formation and that would cooperate with BMP2/4 to spatially restrict *nodal* expression [8].

## Panda, a maternally expressed TGF-beta ligand involved in symmetry breaking

In 2015, we reported the identification of Panda, a maternally expressed TGF-β ligand related to GDF15, Lefty, TGF-β, and Inhibins from vertebrates and to Maverick from flies, that plays a central role in the spatial restriction of *nodal* expression [8]. Panda is necessary for D/V axis specification and embryos lacking Panda show ectopic and unrestricted expression of *nodal* from the initiation of its expression at the 32-cell stage, lasting until late gastrulation and as a result develop as ventralized larvae. Interestingly, this ectopic expression of *nodal* in *panda*

morphants occurs even in the presence of Lefty, suggesting that Lefty is not sufficient for the early restriction of *nodal* expression, and that instead Lefty likely amplifies an initial asymmetry established by Panda. It seems therefore that the reaction–diffusion mechanism involving Nodal and Lefty, although necessary for maintaining the asymmetry of *nodal* expression, is not sufficient to break the radial symmetry of the embryo and to restrict *nodal* early. Several observations support the idea that Panda is a maternal determinant that establishes the D/V axis. First, maternal *panda* transcripts are deposited in a shallow dorsal to ventral gradient in the oocyte, with the presumptive dorsal region showing highest concentration of *panda* transcripts [8]. Second, *panda* transcripts are distributed asymmetrically in the subcortical region of immature oocytes reinforcing the idea that this factor plays an early role in establishment of the D/V axis. Third, local injection of *panda* mRNA into one blastomere of the two-cell stage embryo imposes a dorsal fate to the progeny of the injected blastomere in 100% of injected embryos. Furthermore, consistent with the idea that Panda activity is required locally in the embryo, only local injection of *panda* mRNA into one blastomere of the two-cell embryo, but not injection of *panda* mRNA into the egg, efficiently rescued the D/V polarity of *panda* morphants [8]. This observation is reminiscent of what was observed in the case of Nodal, as Nodal activity too is required locally in the embryo. Only local injection of *nodal* mRNA into one blastomere of 4- or 8-cell stage embryos, but not injection of *nodal* mRNA into the egg, efficiently rescued the D/V axis of embryos previously injected with *nodal* morpholino [4]. Finally, inhibition of *panda* mRNA translation in one blastomere at the two-cell stage causes a cell-autonomous ectopic expression of *nodal* in the progeny of the injected cell suggesting that Panda exerts it repressive effect on *nodal* expression in all cells of the embryo and by a mechanism involving a cell autonomous, i.e., non-diffusible, factor [8].

Since Panda is a member of the TGF-β superfamily and since the loss of Panda appears to synergize with the loss of *bmp2/4* or with the loss of the BMP receptors it was first suggested that Panda might function as a BMP ligand for the BMP receptors Alk1/2 and Alk3/6 and therefore that Panda may antagonize Nodal signaling by promoting BMP signaling, which is known to oppose Nodal signaling [15–19]. However, several observations do not support the idea that Panda promotes BMP signaling. First, Panda does not group with any of the known BMP subfamilies in the phylogenetic studies [8]. Second, overexpression of *panda*, unlike overexpression of *bmp2/4*, does not induce the phosphorylation of pSMAD1/5/8 [8]. Therefore, although Panda is a key regulator of *nodal* expression, since its discovery nearly 10 years ago, the molecular mechanism by which Panda breaks the radial symmetry of the embryo has remained elusive.

In this study, we attempted to solve the intriguing question of the mechanism by which Panda works to break the radial symmetry of the embryo and to promote dorsalization. We found that maternal Panda exerts its effects on D/V axis formation by antagonizing the Nodal and BMP type II receptor ACVRII. First, we analyzed the function of ACVRII in D/V axis specification and found that ACVRII is a limiting factor for Nodal signaling. By using functional studies, we show that Panda antagonizes ACVRII in vivo. Furthermore, we provide evidence that the function of Panda as an antagonist is linked to the presence in its sequence of an atypical ACVRII binding motif and in particular to the presence of a proline residue also highly conserved in the Lefty family of antagonists of Nodal signaling, in place of a conserved serine residue as found in most of the other TGF-β ligands. Finally, through biochemical studies, we show that recombinant Panda secreted in cell culture supernatants binds to ACVRII, and that Panda interacts with the Inhibin co-receptor TBR3 (Betaglycan). Taken together, our results suggest that maternally expressed Panda antagonizes the activity of the Nodal type II receptor ACVRII to break the radial symmetry of the early embryo and specify the D/V axis, and acts in a manner similar to Inhibin α in mammals, which antagonizes Activin signaling by

sequestering ACVRII. We discuss the evolutionary implications of these findings and propose that the original function of Panda may have been in activating the Nodal pathway rather than in its inhibition. These findings suggest a possible scenario concerning the evolution of the mechanisms of D/V axis formation in different classes of echinoderms based on the use of the agonist or antagonist form of Panda.

## Results

### The type II receptor ACVRII is required for D/V axis specification

In vertebrates, Nodal primarily signals through the type I receptors ALK4 and ALK7, and through the type II receptors ACVR2A and ACVR2B [20]. The sea urchin genome encodes a single type I receptor Alk4/5/7, homologous to ALK4, ALK5, and ALK7 from vertebrates, and a type II receptor ACVRII that is homologous to ACVR2, ACVR2A, and ACVR2B from vertebrates [21]. During early development, this constitutively active type II receptor may serve as a common type II receptor for both Nodal and the BMP ligands BMP5/8 and BMP2/4. A recent study showed that, in humans, Nodal can bind with high affinity to BMPRII, a type II receptor primarily used by BMPs, raising the possibility that Nodal may utilize both ACVRII and BMPRII to activate Nodal signaling [22]. To evaluate the role of ACVRII in Nodal signaling, we knocked it down with antisense morpholinos and studied the impact on D/V axis formation of blocking translation of *acvrII* mRNA (Fig 1). Embryos injected with the *acvrII* morpholino lacked any sign of D/V polarity and remained rounded. At the late gastrula stage (24 hours post fertilization (hpf)), the primary mesenchymal cells (PMCs) failed to form bilateral clusters and instead appeared radially distributed around the archenteron (Fig 1A). At the pluteus stage (48 hpf), the injected embryos adopted a bell-like shape, developed with a straight archenteron and numerous ectopic spicules, were excessively pigmented and covered by a thick ciliated ectoderm (Fig 1A). This phenotype is very similar to the phenotype caused by injection of a morpholino targeting the *nodal* transcript, or to the phenotype caused by treatment with the pharmacological inhibitor SB431542, which blocks Alk4/5/7, the Nodal receptor. At the molecular level, embryos injected with the *acvrII* morpholino had lost *nodal* expression at the swimming blastula stage (Fig 1B). The absence of residual Nodal signaling in these embryos, as evidenced by the fully radialized phenotype and by the loss of *nodal* expression in *acvrII* morphants, suggests that ACVRII is most likely the only type II receptor for Nodal, that it is in limiting amount in the embryos and therefore that it is critical for Nodal signaling and D/V axis specification.

### Overexpression of *acvrII* dorsalizes sea urchin embryos and co-expression of *panda* rescues the D/V polarity of *acvrII* overexpressing embryos

Since *acvrII* morphants lack D/V polarity and resemble *nodal* morphants, we expected overexpression of *acvrII* to promote Nodal signaling and to ventralize the embryos. Surprisingly, however, embryos injected with *acvrII* mRNA later displayed a phenotype very different from that resulting from *nodal* overexpression (Fig 1C). At the late gastrula stage (24 hpf), when control embryos formed bilateral clusters of PMCs and showed flattening of the presumptive ventral ectoderm, *acvrII* injected embryos appeared radialized, with the PMCs arranged in a ring around the archenteron (Fig 1C). At the late pluteus stage (72 hpf), these embryos showed ectopic pigmentation, elongated ectopic spicules, and their ectoderm was thin and irregular (Fig 1C and 1D). This phenotype is very similar to the dorsalized phenotype caused by overexpression of *bmp2/4* or to that caused by treatment with recombinant BMP2/4 protein (Fig 1D) [5]. The late dorsalizing action of ACVRII may be explained by the fact that ACVRII is a

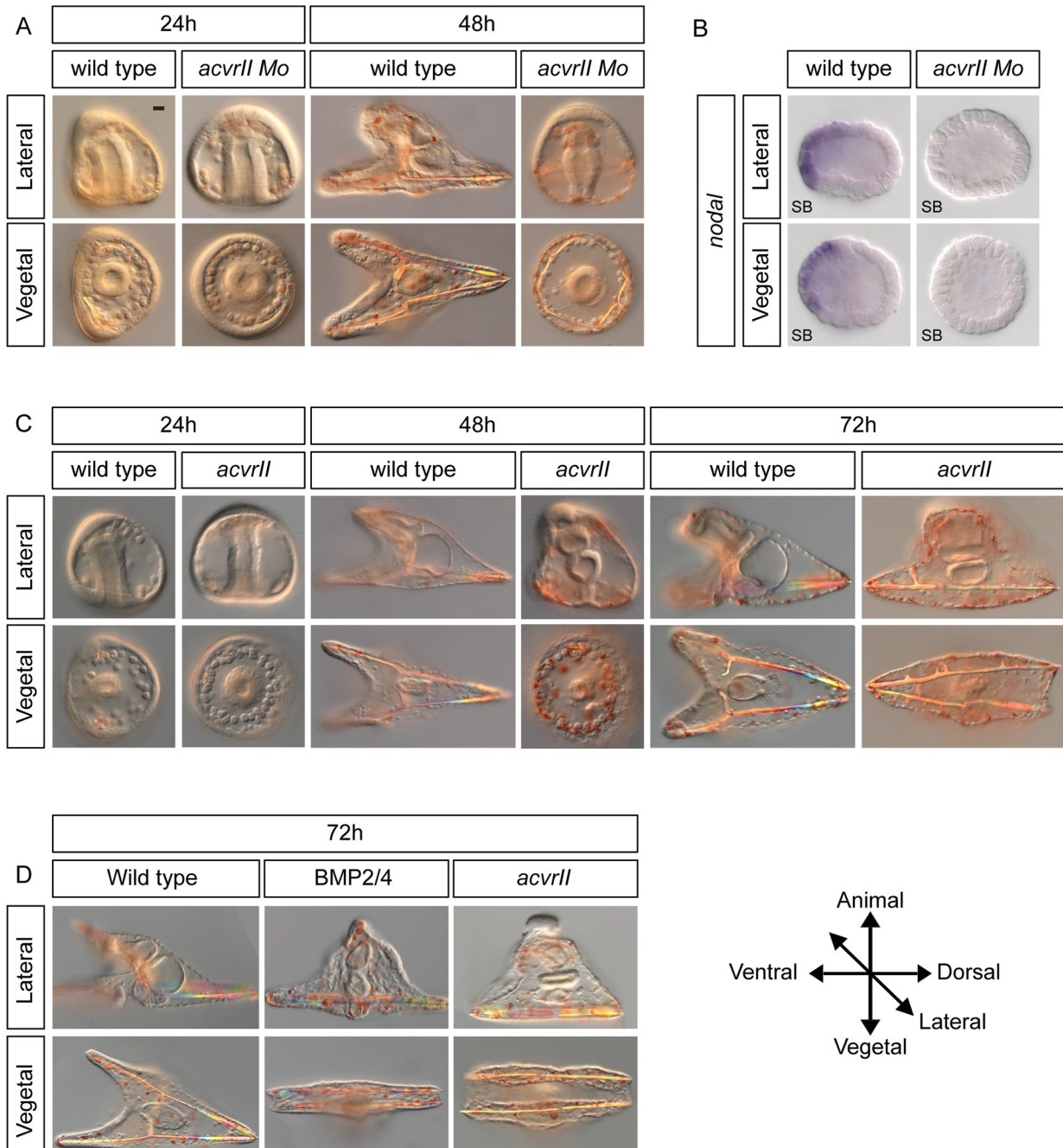

**Fig 1. ACVRII is essential for D/V patterning and overexpression of *acvrII* causes dorsalization.** (A) Morphological phenotypes resulting from injection into the egg of antisense morpholino oligonucleotide targeting the translation start site of the *acvrII* transcript. Knockdown of *acvrII* resulted in the loss of D/V polarity and caused a radialized phenotype, mimicking knockdown of *nodal*. The magnification bar is 10 μm. (B) Expression of *nodal* in wild-type embryos and *acvrII* morphants. *acvrII* morphants exhibited loss of *nodal* expression in the ventral ectoderm. (C) Morphological defects at different developmental stages caused by *acvrII* overexpression. *acvrII* overexpressing embryos appeared radialized at 24 hpf. At 48 hpf and 72 hpf, *acvrII* overexpressing embryos appeared strongly dorsalized, with ectopic pigmentation and presence of elongated spicules supporting a thin and irregular ectoderm. (D) *acvrII* overexpressing embryos adopt a morphology typical of that of embryos dorsalized by treatment with recombinant BMP4 or of embryos overexpressing *bmp2/4*.

constitutively active type II receptor that is shared by both Nodal and BMP ligands. Furthermore, this dorsalized phenotype suggests that the ability of ACVRII to promote BMP signaling when overexpressed in the egg dominates over its ability to activate Nodal signaling [18].

To confirm that *acvrII* overexpression causes dorsalization, we looked at the phosphorylation and nuclearization of Smad1/5/8 at the early blastula stage, when there is no endogenous pSMAD1/5/8 signaling. Indeed, while control embryos lacked pSMAD1/5/8 staining at this stage, embryos injected with *acvrII* mRNA showed strong ectopic activation and nuclearization of pSMAD1/5/8 in all the cells of the ectoderm (Fig 2C). This raised the possibility that this massive and precocious activation of BMP signaling may be the consequence of ectopic expression of *bmp2/4* caused by an increase in the expression of *nodal* following *acvrII* overexpression. Consistent with this idea, embryos overexpressing *acvrII* showed a moderate expansion of *nodal* expression (Fig 2D). However, overexpression of *acvrII* did not cause ectopic expression of the Nodal target gene *bmp2/4* at early or swimming blastula stages ruling out the idea that this ectopic pSmad1/5/8 signaling is caused by a premature expression of *nodal* and/ or *bmp2/4*. Another possibility raised by these results was that the ubiquitous activation of pSMAD1/5/8 at the early blastula stage that followed *acvrII* overexpression reflected the activity of a maternal BMP ligand not yet characterized that would act through ACVRII during early stages, before the induction of *bmp2/4* by Nodal. A good candidate for such a BMP ligand was BMP5/8, since this TGF-beta is expressed maternally and ubiquitously during early development. To test this hypothesis, we co-injected *bmp5/8* MO along with *acvrII* mRNA. ACVRII was able to dorsalize even in the absence of BMP5/8 (S1B Fig). This suggests that the dorsalization caused by *acvrII* overexpression and the associated ectopic Smad1/5/8 activation are likely caused by the ligand-independent activation of BMP signaling by the type II receptor ACVRII [23]. To summarize, overexpression of *acvrII* results in activation of both the Nodal and BMP pathways but the latter has a stronger effect than Nodal and dorsalizes the embryo, possibly by activating BMP/pSmad1/5/8 signaling through the BMP receptors Alk1/2 and Alk3/6.

## Panda antagonizes ACVRII

Previous studies reported that overexpression of *panda* at the egg cell stage does not cause any visible phenotypic effect [8]. This was surprising since reducing the level of Panda by injection of morpholino strongly impacts specification of the dorsal/ventral axis. Since Panda is structurally related to the Inhibins and since Inhibins work by sequestering ACVRII, we re-examined the effect of overexpression of *panda* on Nodal/ACVRII signaling (Fig 2). We injected *panda* mRNA at high concentration and analyzed the expression of *nodal*. We confirmed that overexpression of *panda* at doses up to 1,500 μg/ml does not perturb development. However, at these high doses, overexpression of *panda* significantly reduced the level of expression of *nodal* at the early blastula stage, consistent with the idea that Panda works as an antagonist of Nodal signaling (Fig 2A). This antagonism of Panda on the expression of *nodal* was interesting since, it was the first observable phenotypic effect caused by overexpression of *panda* in the egg. We then compared the effects of injecting into the egg *acvrII* mRNA alone or in combination with *panda* mRNA. Unlike *acvrII* injected embryos, which appeared strongly dorsalized and lacked D/V polarity at the pluteus stage (Fig 2B), a significant proportion of embryos co-injected with *panda* and *acvrII* (56/76; 72%) showed a partial recovery of D/V polarity and proper patterning of the ectoderm and developed with an elongated dorsal apex and ventral arms. This rescue was dependent on the ratios between the concentration of *acvrII* mRNA/ concentration of *panda* mRNA with full rescue at a ratio of 0.6, partial rescue at 0.75, and no rescue at a ratio of 1.75. The antagonism between Panda and ACVRII was confirmed at the molecular level by pSMAD1/5/8 staining. While all embryos injected with *acvrII* mRNA

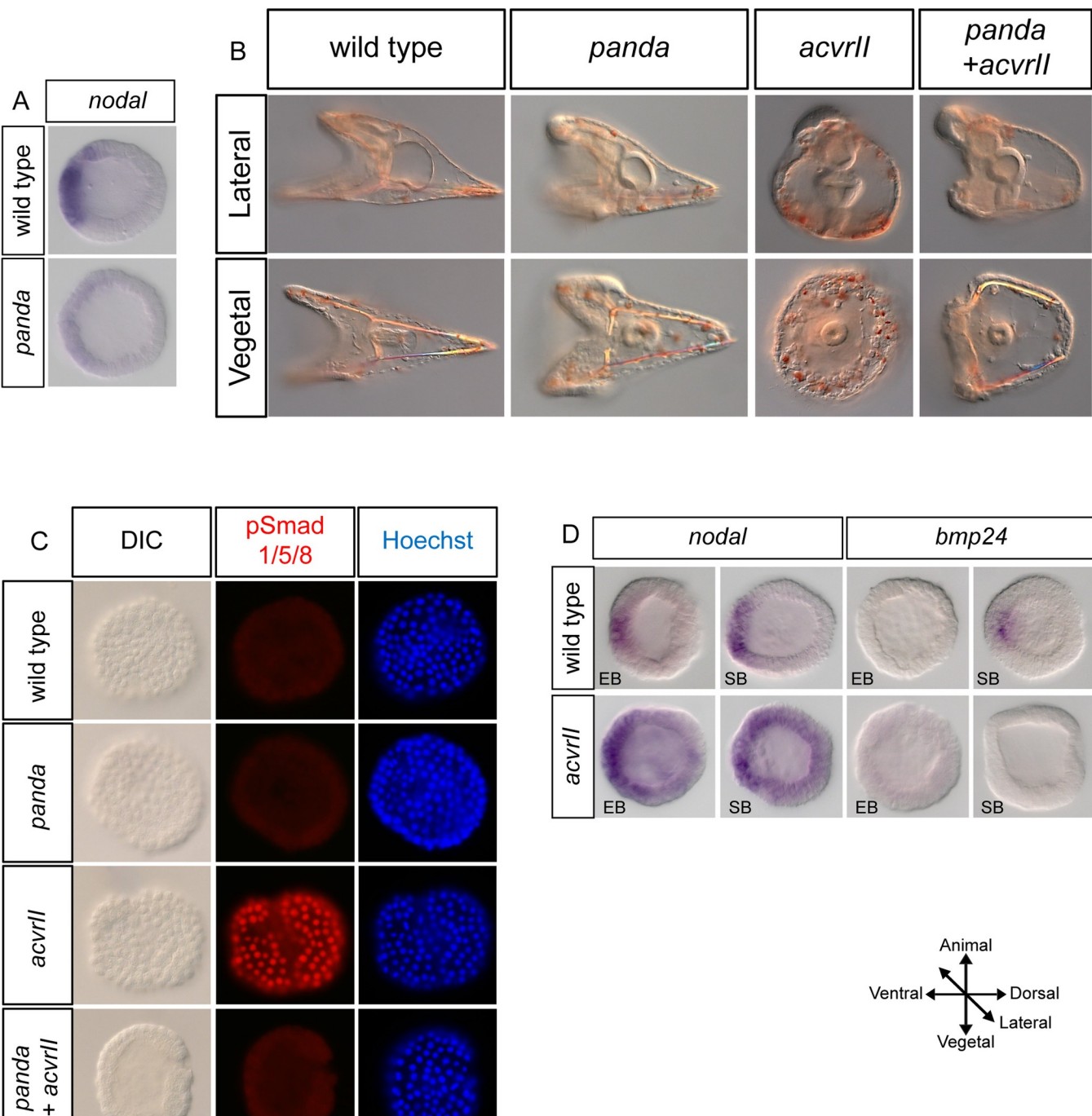

**Fig 2. Co-injection of *panda* mRNA with *acvrII* mRNA suppressed the dorsalization caused by *acvrII* and partially rescued the D/V polarity.** (A) Overexpression of *panda* mRNA (1,500 μg/ml) reduces *nodal* expression at blastula stage. (B) While overexpression of *acvrII* mRNA alone (600 μg/ml) dorsalizes the embryos, co-injection of *panda* mRNA partially rescues normal development and allows the D/V axis to be established. (C) pSmad1/5/8 immunostaining at the very early blastula stage (VEB) in wild-type embryos and embryos overexpressing *panda*, *acvrII*, and *panda+acvrII*. At the VEB stage, control embryos lacked endogenous pSmad1/5/8. Overexpression of *panda* (800 μg/ml) did not activate pSmad1/5/8 signaling. Embryos injected with *acvrII* (600 μg/ml) showed activation and nuclearization of pSmad1/5/8 in all cells at the VEB stage. Co-injection of *panda* mRNA (400 μg/ml) with *acvrII* mRNA (600 μg/ml) blocked the ectopic activation of pSmad1/5/8 caused by *acvrII* overexpression. (D) Overexpression of *acvrII* up-regulates *nodal* expression starting at early blastula stage but it does not cause ectopic expression of *bmp2/4*.

(600 μg/ml) showed ectopic activation of pSMAD1/5/8 at the early blastula stage, embryos co-injected with *panda* and *acrvII* mRNAs (42/61; 69%) did not show any detectable pSMAD1/5/8 signals at this stage and appeared like control embryos (Fig 2C). In contrast, overexpression of *panda* did not antagonize pSMAD1/5/8 signaling induced by overexpression of the constitutively active BMP receptor Alk1/2QD reinforcing the idea that Panda is a Nodal antagonist but that it does not antagonize BMP signaling. (S1A Fig). These findings strongly suggest that Panda antagonizes signaling through ACVRII and that, in vivo, Panda may work predominantly as a Nodal signaling inhibitor.

## Panda blocks the ability of ACVRII to orient the D/V axis and does not require the BMP type receptors Alk1/2 and Alk3/6 to promote dorsal fates

The fact that Panda efficiently orients the D/V axis when injected locally into one blastomere at the two-cell stage and that only local injection of *panda* mRNA, but not global overexpression of *panda*, rescues D/V polarity of *panda* morphants suggests that Panda activity is spatially restricted in the normal embryo and that it is required locally to specify the D/V axis. When injected locally, *panda* mRNA imposes a dorsal fate to the progeny of the injected blastomere in almost 100% of the injected embryos (Table 1).

This dorsal promoting activity is similar to that observed with the local inhibition of Nodal signaling with the *nodal* MO or with the *acrvII* MO [8]. To further characterize how Panda antagonizes the activity of ACVRII, we used an axis-induction assay and tested the effect of local overexpression of *panda* and *acrvII* alone or in combination on orientation of the D/V axis [8]. For this, *acrvII* mRNA along with a lineage tracer was injected into one blastomere of embryos at the two-cell stage and the position of the clone formed by the progeny of the injected cell was recorded at the pluteus stage (Fig 3A). Surprisingly, although global overexpression of *acrvII* efficiently promoted dorsalization, in 100% of the embryos injected locally with *acrvII* mRNA, the boundaries of the clone of *acrvII* overexpressing cells coincided with the ventral regions (ventral ectoderm and ventral endoderm) of the embryo (Fig 3B and 3C). This ventral promoting activity of *acrvII* was clear-cut and similar to that of a constitutively active version of Alk4/5/7 (Alk4/5/7QD), which encodes a constitutively active version of the Nodal receptor Alk4/5/7 (Fig 3D and 3E) [6,24]. Therefore, *acrvII* overexpression has different

**Table 1. Axis induction assays testing the ability of various ligands and receptors to orient the D/V axis following misexpression into one blastomere at the two-cell stage.**

| Condition | Localization of injection clone | | | |
|---|---|---|---|---|
| | Dorsal (%) | Lateral (%) | Ventral (%) | *n* = |
| *panda* mRNA | 95 | 2 | 3 | 505 |
| *acrvII* mRNA | 0 | 5 | 95 | 102 |
| *panda* mRNA + *acrvII* mRNA | 55 | 19 | 26 | 87 |
| *alk4/5/7 QD* mRNA | 0 | 3 | 97 | 74 |
| *panda mRNA + alk4/5/7 QD mRNA* | 0 | 3 | 97 | 67 |
| *panda P461S* mRNA | 23 | 10 | 67 | 88 |
| *panda* mRNA + MO Alk3/6 | 100 | 0 | 0 | 85 |
| *panda* mRNA + MO Alk1/2 | 100 | 0 | 0 | 35 |
| *panda mRNA + MO Alk3/6 + MO Alk1/2* | 95 | 4 | 1 | 56 |
| MO Alk 1/2 | 17 | 36 | 46 | 75 |
| MO Alk 3/6 | 17 | 46 | 36 | 75 |
| *panda P461S* mRNA + *panda mRNA* | 70 | 18 | 12 | 43 |
| *panda P461S+Y426P* mRNA | 78 | 8 | 14 | 37 |

effects when overexpressed globally, when it promotes BMP signaling and dorsal fates, versus locally, when it promotes Nodal signaling and ventral fates in the progeny of the injected cells. The up-regulation of *nodal* and *bmp2/4* expression by *acvrII* in the injected cells likely mimics formation of a *nodal* expressing ventral organizer that produces BMP2/4 and Chordin and promotes their translocation to the opposite uninjected blastomeres which then adopt a dorsal fate in response to BMP signaling [6,24].

Next, we tested the effect of local co-expression of *panda* and *acvrII* mRNAs on D/V axis orientation. Unlike local overexpression of *acvrII* alone, which efficiently imposed a ventral identity to the progeny of the injected cells, local overexpression of *panda + acvrII* failed to impose a ventral identity to the progeny of the injected blastomeres with 55% of the clones adopting a dorsal identity, 26% a ventral identity, and 19% a lateral identity (Fig 3B and 3C and Table 1). Therefore, Panda canceled the ability of ACVRII to orient the D/V axis by promoting ventral fates and significantly reduced the proportion of ventral clones when locally overexpressed with *acvrII*. Furthermore, local co-injection of *panda* mRNA with *alk4/5/7* QD mRNA did not affect the ability of Alk4/5/7 QD to impose a ventral identity to the progeny of the injected cells (Fig 3D and 3E). This suggests that the activity of Panda is required upstream of the Nodal receptors. Finally, we tested the effect of blocking translation of the mRNAs encoding the BMP type I receptors Alk1/2 and Alk3/6 on the ability of *panda* to orient the axis when injected into one blastomere at the two-cell stage. In one preliminary experiment, we had reported that blocking the type I BMP receptors blocked the axis inducing activity of Panda [8]. However, in the course of this study, we repeatedly observed (in 3 independent experiments) that *panda* mRNA efficiently oriented the axis and promoted dorsal fates even after blocking translation of both Alk1/2 and Alk3/6 transcripts (Table 1). Taken together, these results show that Panda blocks the ability of ACVRII to orient the D/V axis, that Panda activity is needed upstream of the Nodal receptors, and that Panda does not require the BMP type I receptors to promote dorsal fates.

## Panda interacts with the Nodal receptors ACVRII and Cripto but not with the BMP receptor Alk3/6

Although the experiments reported above suggested that Panda can antagonize the activity of ACVRII in vivo, it was unknown whether it did so by physically binding to the Nodal receptor complex and in particular to ACVRII. To test if Panda interacts specifically with the Nodal receptor complex, we employed a biochemical approach. Panda, like other TGF-β ligands, is synthesized as a precursor protein with a signal peptide, a pro-domain, and a biologically active mature ligand domain (Fig 4A). We constructed epitope tagged versions of sea urchin Panda, of the type II Nodal receptor ACVRII, and of the Nodal co-receptor Cripto, and these constructs were transfected in HEK293T cells. Upon probing the cell lysates and culture supernatants for production and secretion of Panda by western blot, we noticed that Panda migrated predominantly as a band of 80 kDa under reducing conditions, and that only a faint band was detected at 27 kDa, which is possibly the size of the mature, glycosylated Panda protein. In non-reducing conditions, full-length Panda secreted in the cell culture medium migrated predominantly as high molecular weight bands ranging from 130 to 250 kDa while a mutant form of Panda deficient for dimerization (Panda C451S) migrated predominantly as an 80 kDa band. The size of 80 kDa therefore likely corresponds to the size of the immature monomeric Panda precursor (Panda$^{FL}$), while the smaller less abundant 27 kDa band likely corresponds to glycosylated monomeric proteolytically processed mature form of Panda (Fig 4A). This suggests that the Panda precursor is only partially matured in HEK293T cells. A possible explanation for this incomplete maturation of the Panda precursor is that HEK293T cells do not

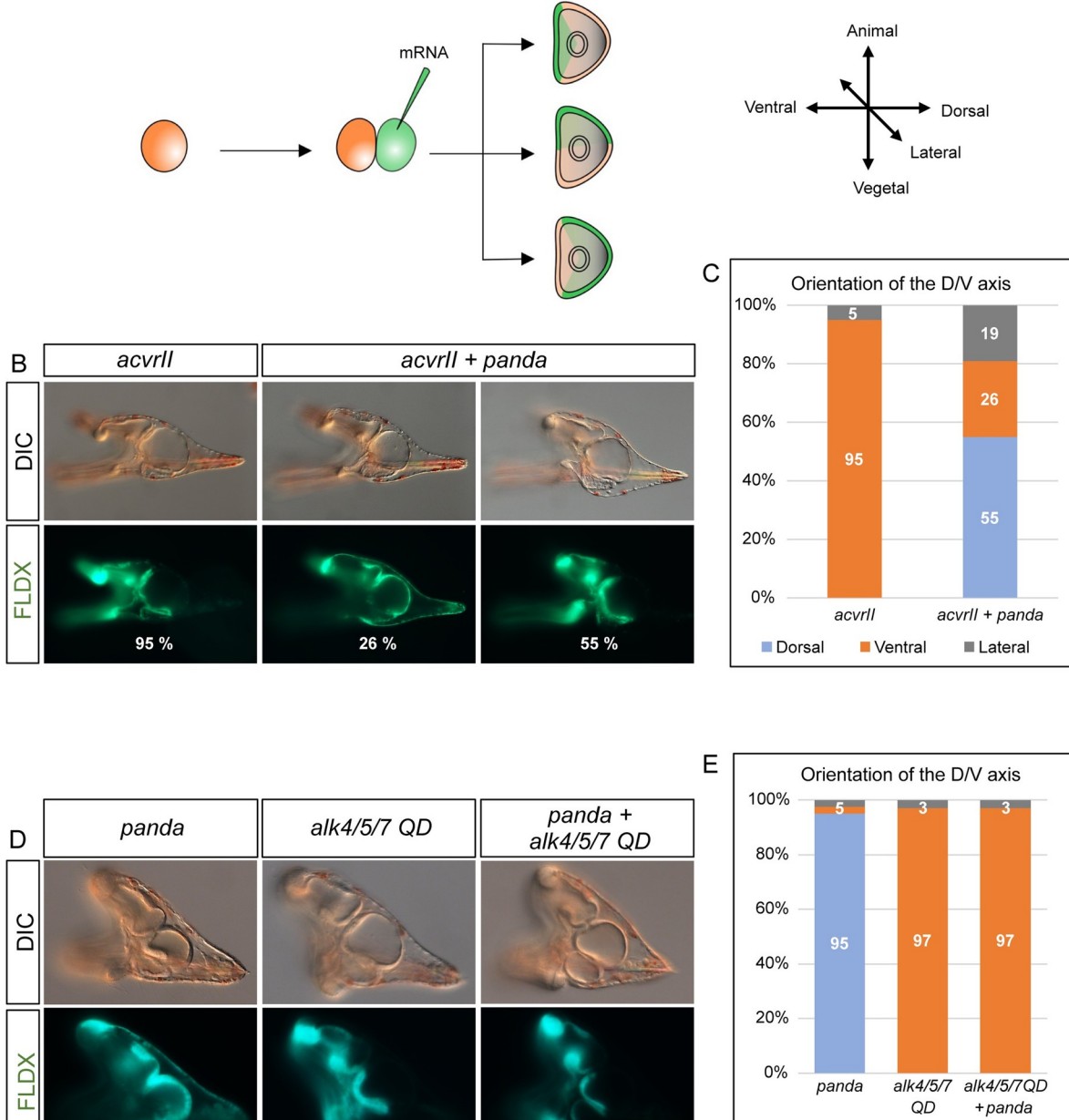

**Fig 3. Panda blocks the ability of ACVRII to orient the D/V axis and Panda acts upstream of the Nodal receptors.** (A) Scheme of the axis specification assay. This assay is based on the fact that In Paracentrotus, there is no correlation between the D/V axis and the first cleavage plane. mRNA encoding the factor to be tested is injected locally into one blastomere at the two-cell stage and the positions of the injection clones is scored at prism/pluteus stages. (B) Effect of local overexpression of *acvrII* mRNA in the presence or absence of *panda* mRNA on the orientation of the D/V axis. Injection of *acvrII* locally into one blastomere at the two-cell stage imposes a ventral identity to the progeny of the injected blastomere in nearly 100% of the injected embryos. Local co-injection of *panda* with *acvrII* blocks the ability of *acvrII* to impose a ventral identity to the clone of injected cells resulting in a mixture of ventral and dorsal clones. (C) Histogram representing the percentage of embryos displaying dorsal, ventral, or lateral injection clones following local overexpression of *acvrII*, or of both *acvrII* and *panda*. The number of embryos scored in each category is indicated. (D) Effect of local overexpression of *alk4/5/7 QD* mRNA in the presence or absence of co-injected *panda* mRNA on the orientation of the D/V axis. Panda does not block the ability of Alk4/5/7 QD to orient ventrally, suggesting that Panda activity is needed upstream of the Nodal receptors. (E) Histogram representing the percentage of embryos displaying dorsal, ventral, or lateral injection clones following local overexpression of *panda*, *alk4/5/7 QD*, or both *panda* and *alk4/5/7 QD*. The number of embryos scored in each category is indicated in Table 1.

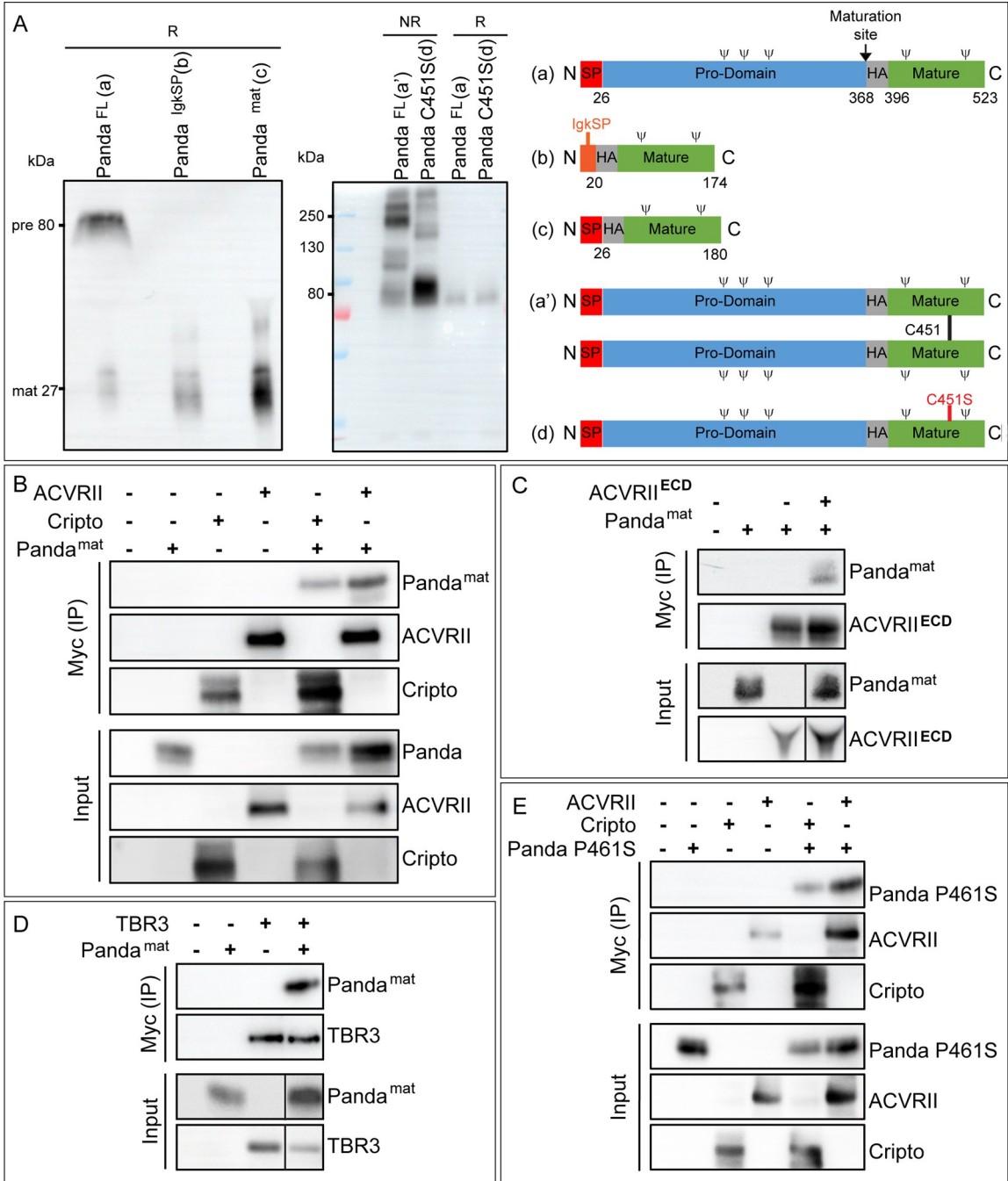

**Fig 4. Structural and biochemical analyses of Panda and Panda protein interaction studies show that Panda can interact with the ACVRII in vitro.** (A) Immunoblot of full-length Panda, Panda^mat, and Panda^IgKSP tagged with 3× HA tags under reducing and non-reducing conditions showing the presence of these proteins in the culture supernatant of HEK293T cells transfected with these constructs. Full-length Panda was secreted predominantly as an 80 kDa precursor form (pre), and to a significantly lesser extent, as a processed 27 kDa mature form (mat). Among the pro-domain deleted versions of Panda, Panda^mat showed a significantly higher secretion into the culture supernatant than Panda^IgKSP. The schemes on the right represent monomeric and dimeric forms of wild-type and mutant Panda molecules. N-linked glycosylation sites are represented by ψ (B) Co-immunoprecipitation of Panda^mat with ACVRII or the Nodal co-receptor Cripto. Cell lysates immunoprecipitated (IP) with anti-Myc (ACVRII or Cripto) in the presence of co-transfected Panda^mat and probed with anti-HA show a 27 kDa band likely corresponding to the glycosylated Panda monomer. (C) Co-immunoprecipitation of Panda^mat with the soluble form of ACVRII in the culture supernatant. Panda^mat binds ACVRII^ECD secreted in the culture medium, suggesting direct binding. (D) Co-immunoprecipitation of Panda^mat with the Inhibin-alpha co-receptor TBR3 (Betaglycan). Cell lysate immunoprecipitated (IP) with anti-Myc (TBR3) in the presence of co-transfected Panda^mat and probed with anti-HA shows a band at the size of the Panda monomer. The lines in C and D indicate that the 2 lanes shown originate from the same

blot but were juxtaposed after cutting an irrelevant lane. (E) Co-immunoprecipitation of Panda[P461Smat] with ACVRII or Cripto, showing that Panda [P461Smat] can form complexes with ACVRII or Cripto.

produce enough of the Furin proteases that are likely required for processing immature Panda. Incomplete processing of TGF-β ligands produced in cell cultures has been documented in numerous studies and in the case of BMPs [25], Nodal [26], and Vg1/GDF1 [27]. To circumvent this problem, we generated a mature version of Panda in which the residues 28–369 corresponding to the pro-domain of Panda were deleted (Fig 4A). We then tested production and secretion of 2 versions of mature Panda: one fused with the native signal peptide (Panda[mat]) and another with the Ig Kappa (Panda[IgKSP]) leader sequence that has been shown to provide a robust secretion of various proteins in mammalian cell cultures. However, we noticed that Panda[mat], which contains the wild-type signal peptide, was produced more abundantly and secreted more efficiently into the culture supernatants than Panda[IgKSP] (Fig 4A). To verify that the biological activity of Panda[mat] was not affected by the pro-domain deletion, we injected mRNA encoding *panda[mat]* locally into one blastomere at the two-cell stage and scored the position along the D/V axis of the clones of injected cells. Just like wild-type Panda, Panda[mat] efficiently oriented the D/V axis, promoting dorsal identities when locally overexpressed, suggesting that deletion of the pro-domain did not alter the biological activity of Panda (S2 Fig). Next, we co-transfected tagged Panda[mat] together with the Nodal receptors ACVRII and Cripto and performed co-immunoprecipitation. Indeed, Panda[mat] co-immunoprecipitated with both ACVRII and Cripto, supporting the idea that Panda may antagonize Nodal signaling by binding to the Nodal receptor complex (Fig 4B). Although Panda co-immunoprecipitated with ACVRII, it was still unclear whether Panda[mat] bound to ACVRII directly or indirectly, i.e., if it interacted directly with ACVRII or with other components of a receptor complex that included ACVRII. To test this, we generated a form of ACVRII known as ACVRII[ECD] that was made of the extracellular domain of this receptor and had the transmembrane region and the intracellular kinase domain deleted and then tested its interaction with Panda[mat] in culture supernatants. Panda[mat] co-immunoprecipated with ACVRII[ECD] in the culture supernatants, suggesting that the interaction between Panda[mat] and ACVRII may be direct (Fig 4C). Finally, in co-IP experiments, Panda failed to co-immunoprecipitate with the BMP type I receptor Alk3/6, reinforcing the idea that Panda is not a BMP ligand and that it does not bind to or signal through Alk3/6 (S3 Fig).

### Panda interacts with the Inhibin co-receptor Betaglycan in vitro

In vertebrates, Inhibins A and B are well-characterized antagonists of ACVRII that function by binding and sequestering ACVR2 through an intermediary co-receptor known as Betaglycan or TbrIII [28]. The sea urchin genome contains a single copy of *betaglycan* [21]. We produced an epitope tagged version of the sea urchin Betaglycan and tested its interaction with Panda[mat]. In addition to ACVRII, Panda interacted specifically with Betaglycan, raising the possibility that Panda might function like Inhibins by sequestering ACVRII and thereby blocking Nodal signaling (Fig 4D).

### Mutation of a single residue of Panda shifts its activity from an antagonist to an agonist of Nodal signaling

The finding that Panda binds to ACVRII raised an intriguing question. If Panda binds to ACVRII, why then does not it cause the recruitment and phosphorylation of the type I Nodal/Activin or BMP receptors leading to activation of Smad signaling? To better understand how

Panda exerts its antagonistic function on the Nodal pathway, we examined its sequence and compared it to the sequences of other ligands of the TGF-ß family. From the analysis of 3D structures, an analogy has been made between the dimeric structures of a TGF-ß ligand sub-units and a pair of hands, with the "wrist" of one hand packing into the palm of the other and with the "fingers" formed by 2 pairs of antiparallel beta sheets extending from the cysteine-knot [29,30]. Ligands that bind to ACVRII and activate signaling such as Nodals, BMPs, and Activins possess a region near the C-terminus that is critical for binding to the type II receptor ACVRII [31–35]. This ACVRII binding region is located in the outer convex curvature of the ß-sheets and is called the knuckle region [30]. We noticed that the sequence of Panda is unusual at the level of this ACVRII binding knuckle region (Figs 5 and S4 and S1 Data). First, while in most of the Nodals, BMPs, and Activins, the ß 6 and ß 7 strands are separated by 4 to 6 amino acids, in both Panda as well as in all Lefty factors, the region separating the ß 6 and ß 7 strands is about twice this size (8–12 residues). Furthermore, while the vast majority of TGF-ß ligands including Nodal, BMPs, GDFs, and Inhibins ß have a conserved serine or threonine embedded in the middle of the ß 6 strand, (in a position equivalent to serine 88 in human BMP2 or serine 90 in human Activin A), 3 remarkable exceptions are the Lefty family of Nodal antagonists, TGF-ß 1 from vertebrates, which does not signal through ACVRII, and human Inhibin α, which, after dimerization with Inhibin β and interaction with Betaglycan, functions as an antagonist of Activin signaling by sequestering ACVRII. Remarkably, unlike the Nodals, GDFs, and BMPs, which possess this highly conserved serine, all the Leftys, whether from echinoderms, hemichordates, or chordates, possess a proline and not a serine in this position. Most interestingly, we noticed that Panda also has a proline in that position, just like all the Leftys, TGF-β1, and human Inhibin α (Figs 5 and S4 and S2 Data). This finding therefore raised the intriguing possibility that the presence of this proline in the ß6 region might be cor-related to the ability of Panda to act as an antagonist of the Nodal/ACVRII pathways.

To test this hypothesis, we mutated the proline 461 of Panda to a serine, generating Pan-da$^{P461S}$, and characterized the effect of this single substitution on the function of Panda in the context of D/V axis specification and *nodal* expression. Unlike Panda, which has no effect on the morphology of the embryos when globally overexpressed, embryos injected with *pan-da$^{P461S}$* showed defects in D/V axis specification starting in gastrulation. At the late gastrula stage (24 hpf), when control embryos acquired an apparent D/V polarity characterized by the bending of the archenteron towards the ventral ectoderm and formation of PMC bilateral clus-ters of PMCs, about 30% of the embryos injected with *panda$^{P461S}$* lacked all signs of D/V polar-ity: they remained rounded and displayed a proboscis in the animal pole region, their archenteron remained straight and surrounded by a radial arrangement of PMCs that later dif-ferentiated into ectopic spicules (Fig 6A). At the pluteus stage (48 hpf), when control embryos had extended along the D/V axis, a fraction of the embryos injected with *panda$^{P461S}$* showed a phenotype that was very similar to that observed following injection of low doses of *nodal* mRNA or of a *panda* morpholino [8], suggesting that these embryos were partially ventralized. qRT-PCR indicated that *panda$^{P461S}$* injected embryos show a 2.7-fold increase in *nodal* expres-sion when compared to control embryos (Fig 6C). We confirmed that this phenotype was due to ventralization by looking at the expression of molecular markers. Unlike control embryos, which showed a well restricted expression of *nodal* and of its target gene *chordin*, 30% of the *panda$^{P461S}$* injected embryos showed ectopic and radial expression of both *nodal* and *chordin* (Fig 6B). To determine if the proline to serine substitution affected the binding of Panda to the Nodal receptors, we checked the interaction between Panda$^{P461Smat}$, the mature form of Pan-da$^{P461S}$ that has the pro-domain deleted, with the Nodal receptors ACVRII and with the core-ceptor Cripto. We observed that Panda$^{P461Smat}$ formed complexes with both ACVRII and Cripto (Fig 4E). Finally, we examined the effect of the P461S mutation on Panda's ability to

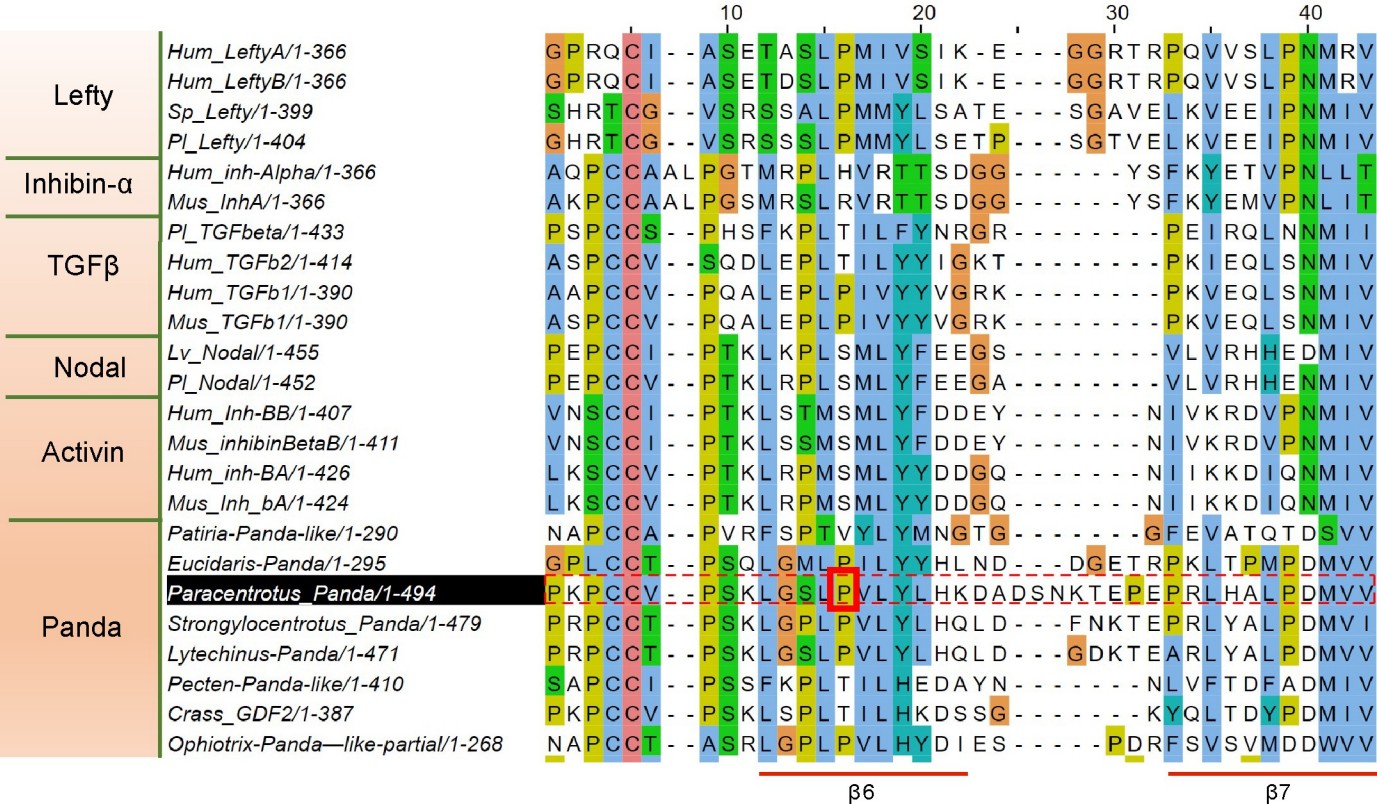

**Fig 5. Structural basis for the antagonistic activity of Panda.** Alignment of the mature domain of Panda with other members of the TGFβ family. Unlike Nodals and Activins that possess a serine in the ß 6 region within the knuckle domain or ACVRII binding region, proteins of the Panda family, like all the Leftys, possess a proline (red asterisk). Also note that while in most of the Nodals, BMPs, and Inhibins ß, the ß 6 and ß 7 strands are separated by 4–6 amino acids, in both Panda and in all the Leftys, as well as in the Inhibin α proteins, the region separating the ß 6 and ß 7 strands is about twice this size (8–12 residues).

orient the D/V axis using an axis orientation assay. While local injection of *panda* mRNA efficiently imposed a dorsal identity in almost 100% of the embryos, local injection of *panda^P461S* mRNA had exactly the opposite effect, now promoting ventral fate in most (71%) embryos, consistent with the view that Panda^P461S promotes Nodal signaling when locally overexpressed (Fig 6E and 6G). One explanation for these results is that the conversion of proline 461 to serine has shifted the function of Panda from an antagonist to an agonist of Nodal signaling. Another explanation, however, is that the Panda mutant protein may work as a dominant negative form of Panda that would inhibit wild-type Panda protein present in the injected blastomere and cause up-regulation of *nodal*. To distinguish between these 2 possibilities, we first tested if the Panda^P461S mutant works as a dominant negative form of Panda. We used the axis induction assay and tested the ability of the Panda P461S mutant to antagonize the axis inducing activity of wild-type Panda. While local overexpression of wild-type Panda efficiently promoted dorsal fates, co-injection of the Panda^P461S mutant and wild-type *panda* did not affect the ability of *panda* mRNA to impose dorsal fates indicating that the Panda^P461S mutant does not work as a dominant negative version (Fig 6F).

We then tried to determine if the Panda P461S mutant synergize with ACVRII as an indication that it may work as an agonist that would promote the assembly of the Nodal receptor complex. In control embryos at late blastula stage, *nodal* was expressed in a restricted domain corresponding to the presumptive ventral ectoderm. Overexpression of *acvrII* (300 μg/ml) weakly increased *nodal* expression while overexpression of the Panda P461S mutant (1,000 μg/

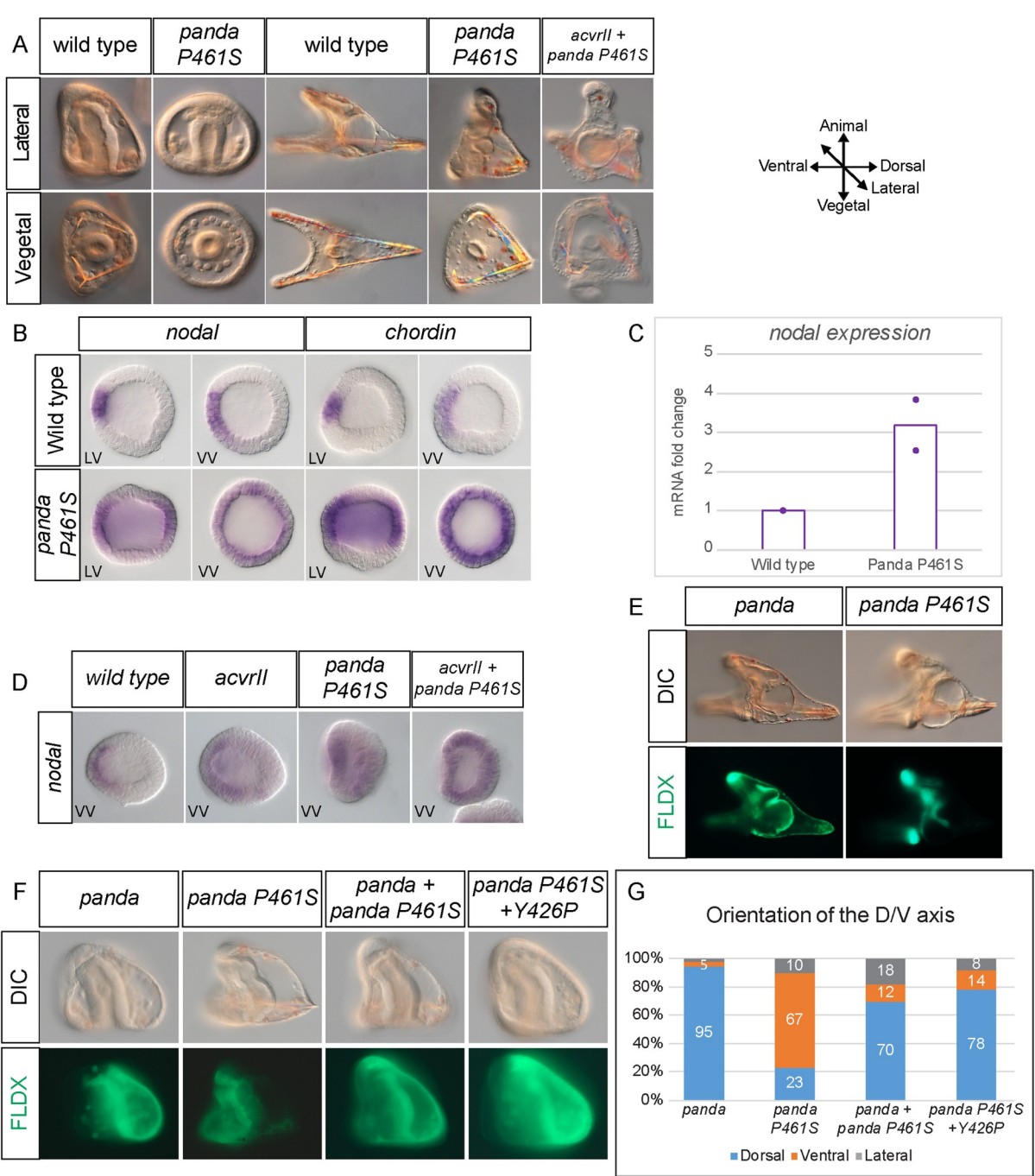

**Fig 6. Morphological phenotypes resulting from the global and local injection of *panda^P461S* mRNA into the egg or two-cell zygotes.** (A) *panda^P461S* overexpressing embryos appear radialized at the late gastrula stage and partially ventralized at the pluteus stage. (B) In situ hybridization for *nodal* and *chordin* in wild-type and *panda^P461S* overexpressing embryos at the late blastula stage. Note that *panda^P461S* injection causes a dramatic ectopic expression of both *nodal* and *chordin*. (C) qRT-PCR analysis of *nodal* transcript levels in wild-type and *panda^P461S* injected embryos at the swimming blastula stage. *panda^P461S* injection results in a 2.7-fold increase in *nodal* expression when compared to wild-type embryos. The values underlying the graph can be found in S1 Data. (D) Synergy between Panda P461S and ACVRII. Co-injection of *panda P461S* mRNA and *acvrII* mRNA result in strong ventralization of the embryos. (E) Effect of local overexpression of *panda* or *panda^P461S* mRNA on D/V axis formation. While local injection of *panda* into one blastomere at the two-cell stage imposes a dorsal identity to the progeny of the injected blastomere, local injection of *panda^P461S* predominantly assigns a ventral identity to the progeny of the injected cell. (F) The *panda* P461S mutant does not work as a dominant negative form of Panda. Introduction of Y426P in the type I receptor binding region of Panda P461S abrogates its ventralizing activity. (G) Histogram representing the percentage of embryos displaying dorsal, ventral, or lateral injection clones following local overexpression of *panda*, *panda* P461S, or *panda* P461S + Y426P.

ml) resulted in 30% of the embryos developing as partially ventralized embryos that displayed a radial expression of *nodal*. Strikingly, co-injection of *panda* P461S mRNA (825 μg/ml) and *acvrII* mRNA (300 μg/ml) resulted in most of the embryos developing as strongly ventralized embryos that showed a radial ectopic expression of *nodal* in the ectoderm (Fig 6A and 6D). This strong synergy between *acvrII* and the P461S mutant further supports the idea that the Panda P461S mutant works as an agonist of Nodal signaling.

Finally, to test if the P461S Panda mutant requires the binding to Alk4/5/7 to signal as an agonist, we introduced in the Panda P461S mutant an additional mutation that should prevent the binding of Panda P461S to the type I receptors. This mutation, Y426 of Panda, is located within the wrist region of the ligand and was identified by mutagenesis as a key residue required for binding of BMP2 to the type I receptor (leucine 51 of hBMP2). Interestingly, in most TGF beta ligands, this position is occupied by a hydrophobic/aromatic residue while this residue is replaced either by a Proline (human and mouse), a positively charged residue (echinoderms, hemichordates, tunicates) or an alanine in Lefty factors that do not bind to Alk4. To test if Panda P461S requires the type I receptor Alk4/5/7 to promote Nodal signaling, we replaced Y426 of Panda by a proline generating the Y426P mutation as the equivalent of the L51P mutation of human BMP2, into Panda P461S and tested its effect in a D/V axis specification assay. While local overexpression of the Panda mutant Panda P461S into one cell at the two-cell stage efficiently imposed a ventral fate to the progeny of the injected cell, addition of the Y426P mutation to the Panda P461S abolished its ability to promote ventral fates and instead efficiently promoted dorsal fates (Fig 6F). Collectively, these results suggest that proline 461 of wild-type Panda is crucial for its function as an antagonist of ACVRII and Nodal signaling. They also suggest that the Panda P461S mutant synergizes with ACVRII to up-regulate Nodal expression and that the ability of the Panda P461S mutant to up-regulate Nodal signaling may require binding to Alk4/5/7.

## Evolutionary origin of the antagonism of Panda on the Nodal pathway

That substitution of a single residue was sufficient to convert Panda from an antagonist of Nodal signaling into an agonist raised the intriguing possibility that the ancestral function of Panda may have been to orient the D/V axis by promoting Nodal signaling and that its function as an antagonist appeared secondarily in the course of evolution following mutation of residue 461 (presumably a serine or threonine) into proline. According to this scenario, the presence of a serine residue in the knuckle region of Panda in the ancestors of echinoderms would have been associated with high affinity binding to ACVRII and with promotion of Nodal signaling and ventral fates during development. The conversion of this serine into a proline in the course of evolution would have secondarily converted the activity of Panda into that of an antagonist of ACVRII and Nodal signaling that would instead promote dorsal fates. To explore further this hypothesis, we did a phylogenetic analysis and we mapped the nature of the residue present at the position equivalent to P461 of Panda onto the different Panda family members (Fig 7). Our phylogenetic analysis, which includes several members of the different echinoderm classes such as the Cidaroids (*Eucidaris tribuloides*), the ancestral urchins and their sister group, the Euechinoidea (*Paracentrotus lividus*, *Strongylocentrotus purpuratus*, *Lytechinus variegatus*), Asteroidea (*Patiria miniata*), Ophiuridea (*Ophiotrix spiculata*) as well as Panda-related sequences from Molluscs (*Crassostrea gigas* and *Pecten maximus*), cephalochordates (*Branchiostoma floridae*), and hemichordates (*Saccoglossus kowaleskii*), is globally consistent with our previous analysis [8] and confirms that Panda is a sea urchin member of a not well-resolved family of TGF-β that comprises both protostome and deuterostome members and that includes GDF15 from vertebrates, Maverick from *Drosophila* and Myostatin-like

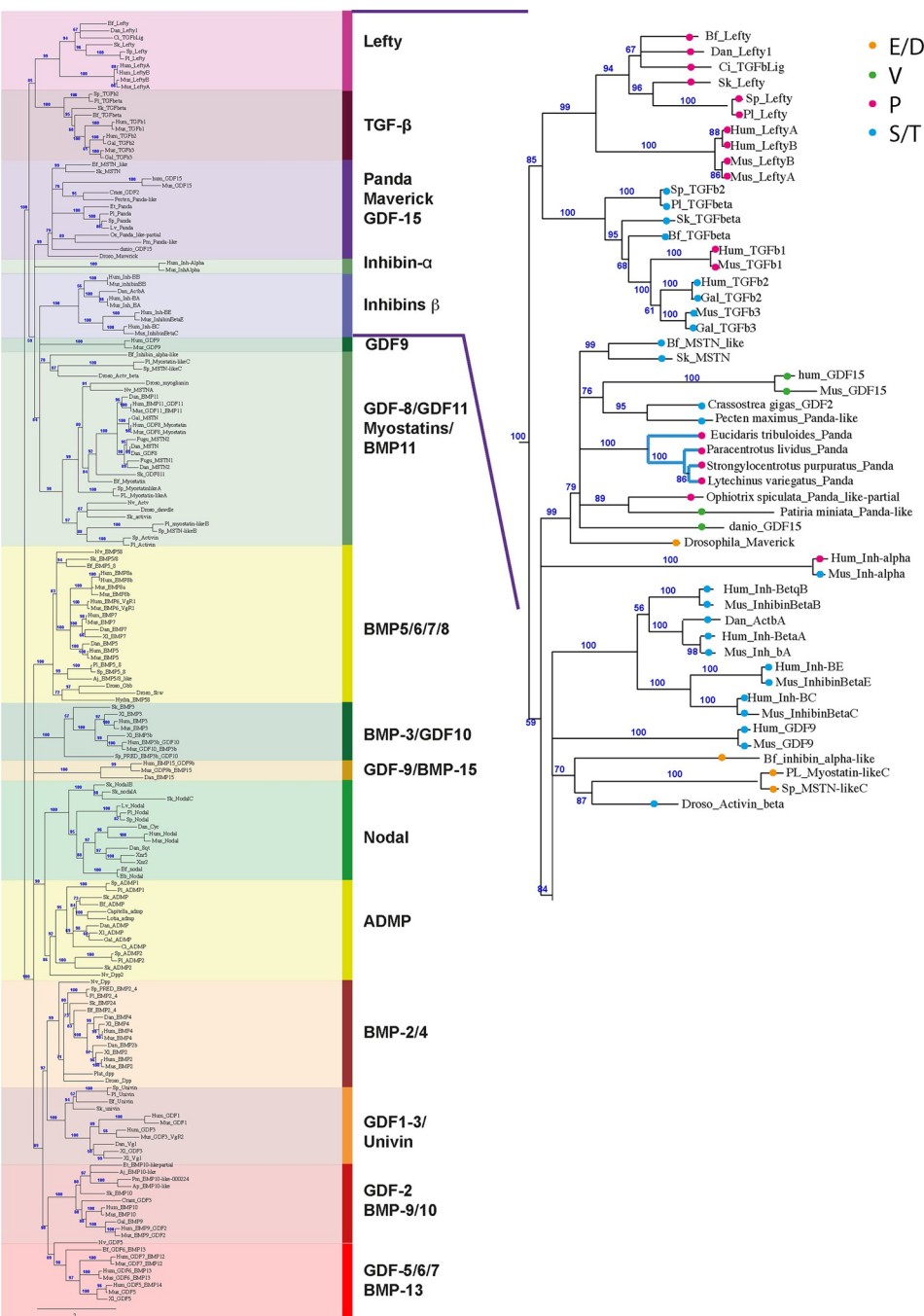

**Fig 7. Phylogenetic analysis of metazoan TGF-β ligands: All Panda sequences from sea urchins, like all Leftys, carry a proline in position equivalent to Panda P461.** Left, maximum likelihood phylogenetic tree of TGF-β sequences from deuterostomes (vertebrates, cephalochordates, hemichordates, tunicates, hemichordates, and echinoderms) and protostomes (arthropods, cnidarians, molluscs, annelids). The full-length sequences of the TGF-β were used for the analysis. The tree was calculated using the maximum likelihood method with PhyML with the substitution model WAG. A consensus tree with a 50% cut off value was derived from the support values of the alRT test. Numbers above nodes represent the approximate likelihood ratio values as percentage of the values supporting the node. The scheme on the right is a close up of the first clades of the tree on the left. The nature of the residue present at the position equivalent to proline 461 of Panda in each species is indicated by a colored dot. The list of the accession numbers of the 182 sequences is provided in the Supporting information (S1 Text).

factors from hemichordates and cephalochordates. Intriguingly, a distinctive feature of Panda proteins from urchins is that the motif W-X-X-W that is highly conserved among TGF-β ligands and that is part of the binding interface with the type I receptor [32,36] is divergent in the Panda family members from sea urchins with only 1 tryptophane being conserved, as in *Eucidaris* (FGFW), in *Lytechinus* (FNW), or in *Strongylocentrotus* (FNW), or with both tryptophanes being replaced by hydrophobic residues as in *Paracentrotus* (LNF). Divergent W-X-X-W motifs are also found in *Paracentrotus* Myostatin-like C (LNG) and in Maverick from *Drosophila* (GFEF). This suggests that the Panda family of antagonists that includes Panda from urchins and Maverick from *Drosophila* may have a reduced ability to bind to the type I receptors.

Previous studies had proposed that the lancelet (*Branchiostoma floridae*) Inhibin/Activin gene is likely to represent an Inhibin α subunit orthologue. Our maximum likelihood phylogeny does not confirm this grouping and shows instead that the Inhibin/Activin protein from the lancelet groups with Activin from *Drosophila* and Myostatin-like C from sea urchin (supported at 70%). This grouping of the lancelet Inhibin α-like with the Myostatin-like C sequences is corroborated by the presence, in the position equivalent to proline 461 of Panda, of a negatively charged amino acid in both Myostatin-like C from urchins and in the Inhibin α-like sequence from *Branchiostoma floridae* and by the lack of the cysteine required for dimerization in both sea urchin Myostatin-like C and in the lancelet Inhibin α-like sequence. We then mapped onto the different members of this family of Panda and Panda-related factors the nature of the residue (present in a position equivalent to P461 of Panda) that appears to direct its activity as an agonist (serine) or antagonist (proline). As shown on Fig 5, all the characterized Panda family members from sea urchins and the single characterized Panda family member from Ophiurids possess a proline, like all the Leftys, like TGF-β1 and like human Inhibin α, and therefore likely work as antagonists, while starfish Panda and all vertebrate GDF15 possess a valine at that position. Only the Panda-like factors from molluscs, hemichordates, and cephalochordates possess a serine at that position, like the vast majority of BMP and Nodal ligands, and likely work as agonists of Nodal/BMP signaling. Since Lefty and TGF-β *sensu stricto* appeared in deuterostomes and since the vast majority of the other families of TGF-β ligands possess a serine or a threonine at that position, it is most likely that the presence of a serine or a threonine in the sequence of Panda at position 461 was the ancestral condition and that in the course of evolution, this serine was replaced by a proline in the Panda sequences from sea urchins, as well as in the Lefty sequences of the common ancestor of deuterostomes and in human TGF-β1 while it was replaced by a valine in the GDF15 and in starfish.

In conclusion, the results presented in this study reveal that Panda exerts its action as an antagonist of the Nodal signaling pathway through its effect on the Nodal type II receptor ACVRII. We showed that Panda antagonizes the activity of ACVRII when it is overexpressed in the egg or when it is locally overexpressed, as in the axis induction assays. Finally, we provided evidence that Panda physically interacts with the Nodal receptors ACVRII and Cripto and that Panda interacts with the Inhibin co-receptor Betaglycan/TBR3 suggesting that Panda may act like Inhibins, by sequestering ACVRII through Betaglycan. Finally, using mutational analysis and functional assays, we have traced back the activity of Panda as an antagonist to the presence of a single proline residue, proline 461, in the sequence of its predicted type II receptor binding motif and that Panda sequences from several sea urchin species appear to possess a proline at that position strongly suggesting that these species rely on a similar antagonism between Nodal/ACVRII and Panda to establish the dorsal-ventral axis. This structure/function analysis of Panda therefore illustrates how mutation of a single residue can dramatically alter the function of a TGF-β resulting in an unprecedented transformation of an antagonist into an agonist of Nodal signaling.

## Discussion

### Panda, a maternal determinant of the D/V axis in the sea urchin embryo but with an enigmatic mechanism of action

The discovery of Panda as a maternally expressed TGF-β ligand responsible for symmetry-breaking and for the spatial restriction of *nodal* expression came as a relative surprise because the high plasticity of the early blastomeres of the sea urchin embryo argued against the existence of maternal determinants that would be spatially restricted and that would direct determination along the D/V axis in the blastomeres that would inherit them. However, Panda clearly fulfills most of the criteria for a maternal determinant of the D/V axis. First, *panda* mRNA is asymmetrically deposited along the D/V axis in the oocyte and early embryos of *P. lividus* (see S9 Fig, [8]). This property distinguishes Panda from other maternally expressed TGF-β ligands such as Univin or BMP5/8, which are ubiquitously expressed in unfertilized eggs and early embryos [6,21]. Furthermore, rescue experiments of Panda knockdown have shown that, just like what had been observed in the case of *nodal*, the activity of Panda must be spatially restricted in the embryo for the rescue experiment to work since only local injection of *panda* mRNA but not injection of *panda* into the egg, can rescue D/V axis formation in *panda* morphants. This observation is reminiscent of what was observed in *Drosophila* for the TGF-β Screw: local expression of the TGF-β *screw*, but not ubiquitous expression, rescues the D/V axis of *screw* mutants [37].

Second, the function of Panda is required very early to restrict *nodal* expression and to establish the D/V axis. Panda morphants are strongly ventralized and do not show any spatially restricted expression of *nodal* before late in gastrulation when they partially recover a D/V polarity. Furthermore, the function of *panda* appears to be required before the reaction–diffusion mechanism that comes into play at the early blastula stage and that relies on the function of Lefty. Third, Panda, like Nodal, also efficiently orients the D/V axis and imposes a D/V identity to cells when overexpressed locally, making it a factor that is both necessary and sufficient to establish the D/V axis. Despite these remarkable properties that make of Panda a very attractive factor to study the process of D/V axis formation, the molecular mechanism by which this factor exerted its functions has remained poorly understood. More specifically, since its discovery, the mechanism by which Panda functions to break the radial symmetry of the embryos and to spatially restrict *nodal* expression has remained enigmatic. Due to the similarity of the loss-of-function phenotypes of Panda and of the BMP type I receptors, it was suggested that Panda may function as a BMP ligand acting through the BMP type I receptors Alk1/2 and Alk3/6. However, phylogenetic analysis indicated that Panda does not group with other members of the BMP superfamily, arguing against the idea that Panda signals through type I BMP receptors. Furthermore, Panda failed to activate pSMAD1/5/8 upon overexpression, a result that definitively ruled out the idea that Panda works like a bona fide BMP ligand but that left unanswered the question of the mechanism by which Panda works.

In this study, by investigating the mechanism by which Panda works, we discovered instead that Panda uses a mechanism that had not been described before for a TGF-β from a non-vertebrate organism: Panda opposes the activity of the Nodal signaling pathway by antagonizing ACVRII, a Nodal and BMP type II receptor. Furthermore, using functional experiments, we have traced back the origin of the antagonism of Panda on Nodal signaling to the presence of a single proline residue in the ACVRII binding region of Panda. This finding allows to draw a comparison of the mechanism by which antagonists of the TGF-β pathway such as Inhibins and Panda work. Finally, using phylogenomics, we have mapped the presence of *panda* orthologues carrying this proline in the sequences of Panda and Panda-like factors in different classes of echinoderms and in different phyla. From this analysis, a surprising conclusion emerges:

The data suggest that the ancestral function of Panda may have been to promote, and not to antagonize, Nodal signaling and that the antagonistic form of Panda possibly emerged in echinoderms, in the last common ancestor of sea urchins and starfish around 250 millions years ago. This analysis suggests that within echinoderms, different evolutionary strategies may have been followed with some classes of echinoderms, i.e., urchins and ophiurids, possibly adopting a mechanism of D/V axis formation based on the spatial restriction of Nodal expression by an antagonist, while other classes (starfish) may have conserved a mechanism based on the spatial entrainment of *nodal* expression by a Nodal agonist.

## Panda as an antagonist of the Nodal type II receptor ACVRII

We have presented several lines of evidence supporting the idea that Panda functions as an antagonist of the Nodal type II receptor ACVRII. First, we showed that overexpression of Panda antagonizes the strong dorsalizing activity of ACVRII that follows the global overexpression of mRNA encoding this factor. When overexpressed into the egg, ACVRII causes the embryos to develop with a dorsalized phenotype, mimicking hyper activation of the BMP pathway. Co-injecting *panda* with *acvrII* mRNAs dramatically blocked the effects caused by overexpression of ACVRII, counteracted its ability to dorsalize the embryos, and partially rescued the D/V polarity. Furthermore, unlike *acvrII* injected embryos, which showed ectopic activation of Smad1/5/8 at the early blastula stage, embryos injected with *panda*+*acvrII* failed to activate Smad1/5/8 at this stage. We showed that Panda also blocked the ability of ACVRII to orient the axis when injected locally into one blastomere at the two-cell stage. Finally, we showed that recombinant mature Panda protein binds to ACVRII and to the Nodal co-receptor Cripto but not to BMP type I receptors. It is important to note, however, that despite the fact that ACVRII can theoretically signal as a type II receptor both for the Nodal and for the BMP pathways, in the blastula stage embryo, Panda may work predominantly as an antagonist of Nodal signaling and not as an antagonist of BMP signaling. This idea is supported by our finding that the phenotype of *acvrII* morphants strongly resembles the Nodal loss-of-function phenotype. Another reason that supports this idea is that, in vertebrates, ACVRII has been shown to act as a low affinity receptor for BMPs [34]. Although we do not know if this also the case in the sea urchin, it is possible that BMP2/4, which is the main player that participates in D/V axis specification in this organism, also uses ACVRII as a low affinity type II BMP receptor. ACVRII may therefore be a limiting factor for Nodal signaling, as it is the only type II receptor for Nodal, while BMPs can utilize BMPRII in addition to ACVRII. ACVRII overexpression likely causes dorsalization and activation of Smad1/5/8 due to ligand-independent hyperactivation of the 2 BMP type I receptors Alk1/2 and Alk3/6. Consistent with the idea that Panda antagonizes Nodal signaling by antagonizing ACVRII, we found that Panda also binds to the extracellular domain of ACVRII. The key observation that Panda does not bind to the BMP receptor Alk3/6 further supports the idea that Panda is not a prototypical BMP ligand. Interestingly, Panda was also found capable to complex with TBR3 (Betaglycan), the co-receptor for Inhibins α in vertebrates. However, further studies are needed study the role of TBR3 in D/V axis specification in the sea urchin.

Finally, the finding that Panda antagonizes ACVRII suggests an explanation for an intriguing observation made following local injection of the *panda* morpholino: Unlike injections of the *alk3/6* or *alk1/2* morpholinos, which promote ventral fates but do not cause ectopic expression of *nodal* in all the progeny of the injected cell, injection of the *panda* morpholino promotes ventral fates and causes the cell autonomous de-repression of *nodal* expression, i.e., causes the ectopic expression of *nodal* in all the progeny of the injected cell [8]. A possible explanation for this effect of the *panda* morpholino is that blocking translation of *panda*

mRNA frees ACVRII and possibly also Cripto, within the blastomere that receives the morpholino making it available for Nodal signaling, which in turn causes unrestricted, Lefty-independent, *nodal* autoactivation in the progeny of the injected cell.

Collectively, the results obtained in this study improve our knowledge of the mechanism of symmetry breaking and of spatial restriction of *nodal* expression by Panda. Panda antagonizes ACVRII during the early stages, likely resulting in reduced Nodal autoregulation on the dorsal side, where Panda is initially expressed (Fig 8). This would result in a gradient of Nodal autoregulation along the D/V axis that ultimately would lead to the spatial restriction of *nodal* expression to the presumptive ventral side. At later stages, Nodal induces the expression of *lefty*, which maintains the initial asymmetry in *nodal* expression created by Panda.

## Molecular mechanism of the antagonism of Panda on the Nodal signaling pathway

The example of Panda shows that the nature of a single residue in the sequence of a TGF-β can determine its agonistic or antagonistic function in the pathway that it normally regulates. This is the case for Proline 461 of Panda, which is the functional equivalent of serine 88 of BMP2

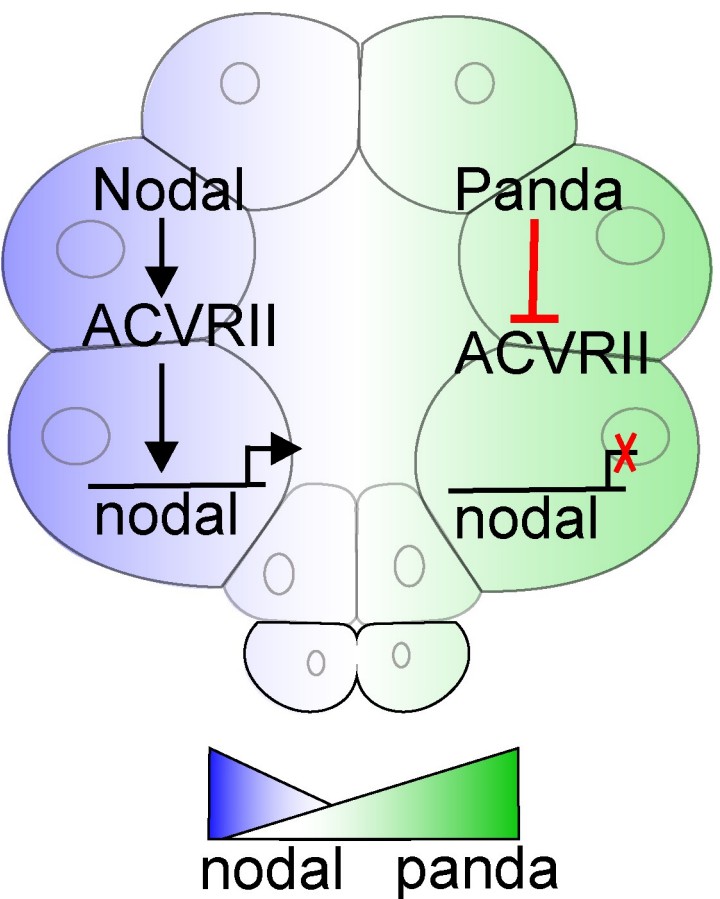

**Fig 8. Model of symmetry breaking during D/V axis specification.** Nodal is ubiquitously induced at the 32-cell stage while maternal *panda* mRNA is enriched in the presumptive dorsal territory. Translation of *panda* mRNA generates a broad gradient of Panda protein along the D/V axis. Panda antagonizes the Nodal type II receptor ACVRII, likely resulting in an asymmetry of Nodal signaling along the D/V axis. Nodal is unable to auto-stimulate its own expression in the dorsal territory and as a result, expression of *nodal* is restricted to the presumptive ventral territory. During later stages, this asymmetry is maintained by Nodal/Lefty reaction–diffusion mechanism.

[34,38]. Structural studies have shown that the "knuckle" type II receptor binding site of BMPs and Activins has a horseshoe shape and is largely hydrophobic. In this knuckle domain, residues that determine the affinity for type II receptor binding have been identified by mutagenesis and by protein interaction studies [32]. Among these residues, a serine that occupies a central position in the horseshoe-shape binding site is extremely conserved. This serine (serine 88 in human BMP2) is found at the equivalent position in the vast majority of ligands that signal through Activin receptors and it has been shown to be important for the function of these ligands [34,38]. This serine residue is engaged in a hydrogen bond with a leucine residue (L61) of ACVRII. Suppression of this hydrogen bond, like in the ACVRII L61P mutant, results in a drastic (2,500-fold) reduction of the affinities of Activin, but not of BMP2 or BMP7, for ACVRII [34]. Panda, unlike Activin or BMPs, possesses a Proline instead of the conserved serine in the β 6 region of the knuckle domain, and its function as an antagonist critically requires this proline residue. Based on the work on human Activin, it is reasonable to assume that the presence of a proline in place of a serine prevents formation of the crucial hydrogen bond between Panda and ACVRII, which in turn may drastically reduce the affinity between Panda and ACVRII. Alternatively, due to its cyclic structure, Proline may disrupt secondary structures such as beta sheets and alpha helices. It is therefore possible that the presence of a proline in the β6 region of Panda disrupts the secondary structure of this region of Panda and of the knuckle region involved in binding of ACVRII. However, why reducing the affinity of Panda for ACVRII would confer to this factor the activity of an antagonist of the Nodal pathway is not well explained by this mechanism. In addition to ACVRII, Panda may target other components of the Nodal receptor complex reinforcing the antagonism of Panda on the Nodal pathway. In particular, Panda may act in part like Lefty, by binding to and sequestering the Nodal co-receptor Cripto. The finding that Panda co-immunoprecipitates with Cripto is consistent with this idea and the presence of a Proline in the knuckle region of both Panda and all the Lefty factors further reinforces this hypothesis. A third possibility is that Panda binds with a high affinity to ACVRII but that proline 461 renders ACVRII unable to phosphorylate the type I receptor Alk4/5/7. The key observations that Panda antagonizes the early Smad1/5/8 activation caused by global overexpression of ACVRII or the ventralization caused by local expression of ACVRII and that Panda co-immunoprecipitates with ACVRII strongly suggest that Panda binds directly to ACVRII and down-regulates its activity preventing the recruitment and the phosphorylation of the BMP or Nodal type I receptors. Finally, an attractive possibility that would be supported by studies on the role of serine 88 in binding of Activin to ACVRII and by our functional data is that Panda may work like the α chain of Inhibins, by preventing signaling from Nodal-Panda heterodimers (Fig 9). The antagonism in this case would result from the asymmetric properties of the Nodal-Panda heterodimer. In particular, the antagonism would result from the high affinity association of Nodal with a single ACVRII type II receptor and through the association of Panda with the membrane proteoglycan Betaglycan co-receptor. Such a complex, made of ACVRII-Nodal-Panda-Betaglycan would not be functional in term of signaling since it would lack the type I receptor Alk4/5/7 and since it would contain a single ACVRII molecule. Therefore, it would inhibit Nodal signaling by sequestering ACVRII. According to this hypothesis, the Nodal-Panda heterodimers would play an antagonistic role that would functionally be the opposite of that played by the Nodal-Univin heterodimers that mediate strong Nodal signaling [39]. This hypothesis would also be consistent with the results on the serine 88 mutant of human Activin and with the idea that the presence of a proline in position 461 of Panda is not compatible with high affinity binding of Panda to ACVRII [34,38] and, importantly, it would also be consistent with our finding that Panda binds to Betaglycan. Again, biochemical experiments to determine the composition of the Panda protein dimers as well as functional analysis of *betaglycan* may provide an answer to that question.

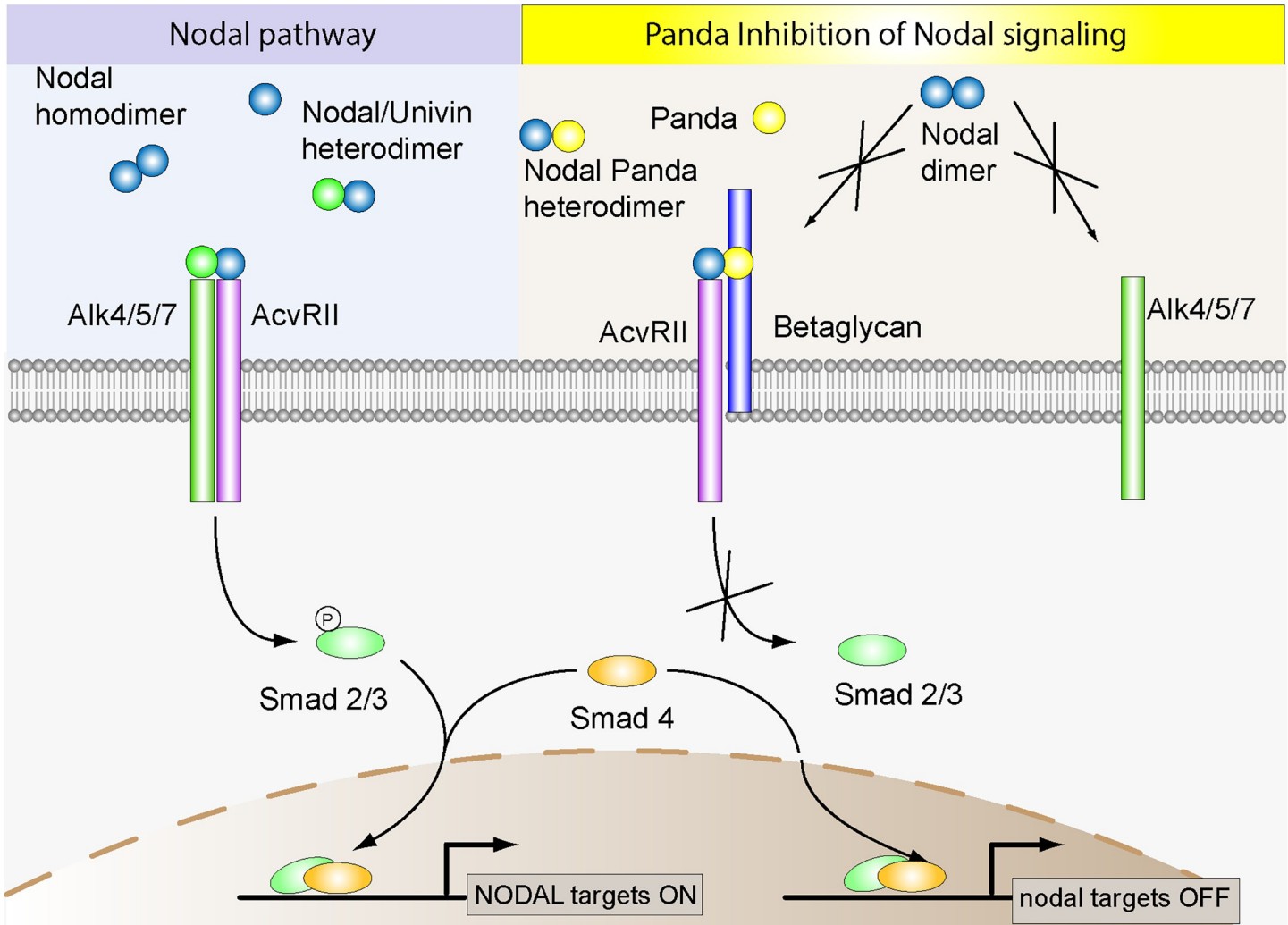

**Fig 9. Proposed model of the mechanism by which Panda antagonizes Nodal signaling.** Panda is proposed to antagonize Nodal signaling by forming asymmetrical heterodimers with Nodal. In the Panda-Nodal heterodimer, Nodal can interact with high affinity with ACVRII while Panda cannot do so due to the presence of proline 461 in its ACVRII binding motif. In contrast, Panda interacts with high affinity with Betaglycan. This complex, made of ACVRII-Nodal-Panda-Betaglycan, cannot signal due to the absence of the type I receptor in the receptor complex and therefore inhibits Nodal signaling by sequestering ACVRII. Note that Panda may also form complexes with other factors such as Cripto or Alk4/5/7 and these complexes may contribute to the antagonism of Panda of Nodal signaling. Also note that the Nodal-Panda heterodimer, which attenuates Nodal signaling, would play a role opposite to that of the Nodal-Univin heterodimer, which promotes strong Nodal signaling.

### Evolutionary origin of the antagonism of Panda on the Nodal pathway: Possible ancestral function of Panda as a Nodal pathway agonist predicted from the P461S mutant

The most striking finding of this study is that substitution of a single residue of the knuckle domain of Panda converts this factor from an antagonist to an agonist. Panda[P461S] has the opposite effect of wild-type Panda when overexpressed locally and promotes ventral fates. While wild-type Panda promotes dorsalization, Panda[P461S] injected embryos are partially ventralized and exhibit ectopic expression of the ventral marker gene *nodal* and of its target gene *chordin*. That mutation of a single residue of Panda changes radically the nature of this TGF-β ligand raises the intriguing possibility that the ancestral function of Panda may have been to activate ACVRII and Nodal signaling. A possible scenario is therefore that in the course of

evolution, a version of Panda containing a serine in the β6 region was used as a maternal activator of Nodal signaling to specify the D/V axis. During evolution, replacement of this serine (serine 461) by a proline might have switched the role of Panda from an agonist to an antagonist of the Nodal pathway. While the agonist form of Panda may have promoted Nodal signaling and entrained the Nodal autoregulatory loop on the presumptive ventral side, the antagonist form of Panda, would have also broken the radial symmetry of the embryo and oriented the D/V axis but by promoting dorsal fates and by a different mechanism based on interruption of the Nodal autoregulatory loop.

Finally, in this study, we have also attempted to trace back the emergence of Panda as an antagonist of the Nodal pathway in the course of evolution. To delineate the emergence of the antagonistic form of Panda, we have mapped the nature of the residue present in the β6 region in the position equivalent to proline 461 of Panda onto a phylogenetic tree of a comprehensive set of TGF-β sequences. We also examined the presence of an insertion between the β6 and β7 beta strands and of a modified W-X-X-W motif since these appear to be characteristic features of Panda factors. This analysis revealed that the antagonistic forms of Panda, i.e., carrying a proline in the position equivalent to proline 461 of Panda from *P. lividus* is present in the genome of the pencil urchins Cidaroids, considered to be ancestral to the other modern urchins, and in the genome of Ophiurids. Although *panda-like* sequences are present in the genomes of lophotrocozoa (molluscs) and ecdysozoa (insects) as well as in hemichordates and cephalochordates, no clear functional equivalent of Panda (i.e., with a proline in the β6 region, an insertion between the β6 and β7 beta strands and a modified W-X-X-W motif) could be found in these clades. The data are therefore consistent with the idea that Panda as an antagonist emerged in the last common ancestor of Cidaroids and Euechinoids, which are thought to have diverged 268 millions years ago [40]. The presence of a Panda-like gene in *ophiotrix spiculata* (Ophiurids) containing a proline in the β6 region but lacking the insertion between β6 and β7 and without a modified W-X-X-W motif suggests that the emergence of Panda as an antagonist may be even more ancient and may have occurred in the last common ancestor of starfish and sea urchins (eleutherozoans). It is possible that in different echinoderms, Panda may still possess a serine instead of a proline in position 461, and that these serine variants may potentiate instead of antagonize Nodal signaling. As new genome sequences and transcriptomes become available, future studies should investigate this possibility in order to map which clades of echinoderms possess a proline in the β6 region and may use the antagonistic form of Panda to specify the D/V axis and which may have retained a serine in the β6 region and may use the agonist form of Panda to entrain *nodal* expression. These studies should help to better understand the evolution of the mechanisms of D/V axis specification in the course of evolution.

While our data suggest that Panda may have shifted from an agonist to an antagonist, a similar evolutionary scenario based on the conversion of an agonist to an antagonist in the course of evolution has been proposed to explain the evolutionary history and the emergence of the antagonistic property of the α chain of Inhibins [41]. During the evolution of vertebrates, following a duplication event, the β-subunit of Inhibins is thought to have undergone a series of changes to evolve into a functional α-subunit. Among these changes, the loss of a β-subunit helix α3 region required for binding to the type I activin receptors due to insertion of a proline-rich region was proposed to be a key event that shifted the Inhibin β ligand from an agonist to an antagonist of Activin signaling in mammals [41]. Although the evolutionary scenarios that converted Panda and Inhibin β from an agonist to an antagonist may appear superficially similar, it is important to note that the mechanisms that have driven these evolutionary trajectories are likely to be different. In particular, in the case of Inhibin α, a deletion of the wrist region seems to have directed the shift from an agonist to an antagonist [41], while

in the case of Panda, a mutation of the knuckle region or ACVRII binding motif of Panda seems to be responsible for this shift. The example of Panda, showing that mutation of a single residue of the ACVRII binding motif in the course of evolution was responsible for conversion of an agonist into an antagonist, is therefore particularly striking and unique in the field of TGF-β signaling. Interestingly, the sequence of human Inhibin α, also contains a proline residue in the position equivalent to proline 461 of Panda. This raises the possibility that in addition to deletion of the wrist region, a mutation within the knuckle region of Inhibin β that converted a serine into proline may have contributed to the shift from agonist to antagonistic and to the emergence of human Inhibin α. It would therefore be interesting to test if the presence of this proline determines, like in the sea urchin, the antagonistic activity of human Inhibin α. Similarly, it would be interesting to test the effect of mutating serine 336 of mouse Inhibin β and to determine if introduction of this proline residue increases the potency of this protein to antagonize Activin signaling.

Members of the TGF-β superfamily such as TGF-β sensu stricto, Activins, GDF15, or Myostatins are implicated in a number of pathogeneses and metabolic diseases such as diabetes [42], obesity, cachexia, and cancer [43,44]. The development and generation of variants of these TGF-β with antagonistic properties is therefore of a high medical interest. Panda offers an interesting example of an endogenous antagonist of Nodal signaling and understanding the molecular basis of this antagonism should be relevant to studies on TGF-β signaling in vertebrates since it may help to better understand the evolution and the regulation of the Nodal/GDF/Activin/TGF-β signaling pathways that are important both during embryogenesis and during homeostasis in the adult.

## Materials and methods

### Animals, embryos, and treatments

Adult sea urchins (*P. lividus*) were collected in the bay of Villefranche-sur-Mer. Embryos were cultured as described in Lepage and Gache (1989, 1990). For in situ hybridization, fertilization envelopes were removed by adding 1mM 3-amino-1,2,4 triazole (ATA) 1 min before insemination to prevent hardening of this envelope followed by filtration through a 75 μm nylon net.

### Phylogenetic analysis

TGF-β sequences from deuterostomes (vertebrates, cephalochordates, hemichordates, tunicates, hemichordates, and echinoderms) and protostomes (arthropods, cnidarians, molluscs, annelids) were recovered from Genebank (http://www.ncbi.nlm.nih.gov/) using well-characterized orthologs of each TGF-β family member from human or mouse. The list of accession numbers of the 182 sequences is provided in the Supporting information (S1 Text). Full-length sequences were aligned using ClustalOmega with default parameters (http://www.ebi.ac.uk/Tools/msa/clustalo/), gaps were removed from the alignment. The full complement of TGF-β sequences was recovered and used in the analysis in the case of human, mouse, sea urchin, *Saccoglossus*, *Branchiostoma*, and *Drosophila*. However, only a subset of sequences from, *Gallus*, *Xenopus*, *Danio*, *Ciona*, *Crassostrea*, *Platynereis*, *Hydra*, and *Nematostella* were included in the analysis.

The analysis was performed on the Phylogeny.fr platform [45] and comprised the following steps. The phylogenetic tree was reconstructed using the maximum likelihood method implemented in the PhyML program (v3.1/3.0 aLRT). The default substitution model was selected assuming an estimated proportion of invariant sites (of 0.006) and 4 gamma-distributed rate categories to account for rate heterogeneity across sites. The gamma shape parameter was estimated directly from the data (gamma = 1.971). Reliability for internal branch was assessed

using the approximate likelihood ratio test (aLRT) test (SH-Like). Graphical representation and edition of the phylogenetic tree were performed with TreeDyn (v198.3).

## Cloning of cDNAs and construction of epitope tagged versions of Panda, BMP, and Nodal receptors

*tbr3* ORF was amplified by RT-PCR from RNA extracted from pluteus larvae and cloned into pCS2 using NEBuilder HiFi DNA Assembly Master Mix with the following oligos:

TBR3 pCS2 overlap Fw–
5′ CAGGATCCCATCGATACCATGACCTTTGACCTTAGAGATC 3′
TBR3 pCS2 overlap rev–
5′ GACTCACTATAGTTCTAGATCATACCGATTCCCCAATGG 3′
pCS2 XbaI Fw–
5′ TCTAGAACTATAGTGAGT CGTATTACG 3′
pCS2 XbaI rev–
5′ GGTATCGATGGGATCCTGC 3′

Pro domain deleted versions of Panda and Panda[P461S] and the mature form of Panda with a signal peptide from the IgK chain (Panda[matIgKSP]) were cloned using the following oligonucleotides:

Panda-mat Fw–5′ CAGAGTGGGGTAAGCAATGGGG 3′
Panda SP rev–5′ CGCGGCGACGTGTGGGAGGGTG 3′
Panda IgKSP rev–
5′ ACCAGTGGAACCTGGAACCCAGAGCAGTACCCATAGCAGGAGTGTGTCTG
TCTCCATGGTATCGATGGGATCCTGC 3′

The extracellular domain of ACVRII was cloned using the flowing oligos using pCS2 ACV-RII as template

ACVRII ECD rev–5′ ACTCTCATTCTTCCTTTTCTTCTCC 3′
ACVRII ECD fw–5′ GGGGGTGGCGGACATCGATTTAAAGCTATGGAGC 3′

Two copies of the Myc epitope tag were added to TBR3 using pCS2 TBR3 as template with the following oligos:

TBR3 2Myc fw–
5′ GGACCCATTGGGGAATC GGTAATGGAGCAAAAGC TCATTTCTGAAGAGGAC
TTGAATGAAATGGAGCA AAAGCTCATTTCTGAAGA GGACTTGAATGAATAG 3′
TBR3 before Myc rev–5′ GTTCATAAGTCCCCTTGCAGG 3′

3HA tags were introduced into the cDNA sequences of Panda[mat] and Panda[P461Smat] using the following oligonucleotides, which were used to amplify pCS2-Panda[mat] and pCS2-Panda[P4161Smat], respectively. The resulting PCR product was phosphorylated and re-ligated.

## qPCR

QPCR was performed as described previously (Range and colleagues) on a StepOne instrument. cyclin-T was used as a reference gene [46]. RNA was extracted from injected embryos using Trizol and treated with DNaseI before reverse transcription. cDNA synthesis was performed using a mixture of random and anchored oligo-dT20 primers. Uninjected embryos were used as control. Delta Ct = Ct target gene—Ct reference gene. Fold change = $2^{-(\text{delta Ct experimental sample-delta Ct calibrator sample})}$.

qPCR oligonucleotides used:
*nodal qPCR rev*–5′– CTC GA GTT CAG CAA GAT GG—3′
*nodal qPCR fw*–5′–TTC TAA ACG GGA GTG CAA GG—3′

## Immunostaining

Embryos were fixed with 4% formaldehyde for 15 min, and then briefly permeabilized with methanol. Anti-Phospho-Smad1/5(S463/465)/9(S465/467) was used at 1/200. Embryos were imaged with an Axio Imager M2 microscope.

## Overexpression of mRNAs and morpholino injections

For overexpression studies, the coding sequence of the genes analyzed was amplified by PCR with a high-fidelity DNA polymerase using oligonucleotides containing restriction sites and cloned into pCS2. Capped mRNAs were synthesized from NotI-linearized templates using mMessage mMachine kit (Ambion). After synthesis, capped RNAs were purified on Sephadex G50 columns and quantitated by spectrophotometry. RNAs were mixed with Rhodamine Lysine-Fixable Dextran (RLDX) (10,000 MW) for phenotypic analysis or with Fluoresceinated Lysine-Fixable Dextran (FLDX) (70,000 MW) for cell lineage analysis at 5 mg/ml and injected in the concentration range of 100 to 1,500 µg/ml.

Wild-type *panda* mRNA was injected at concentrations ranging from 500 µg/ml to 1,500 µg/ml. *panda*$^{P461S}$ mRNA and Panda$^{P461S}$ +Y426P were injected between 1,200 and 1,500 µg/ml. *acvrII* mRNA was injected at 400 µg/ml for one-cell stage injections and at 100 µg/ml for two-cell stage injections. *alk4/5/7 QD* mRNA was injected at 500 µg/ml [5].

The *acvrII* and *bmp5/8* translation blocking morpholinos were described previously [8]. The *acvrII* morpholino oligonucleotide was dissolved in sterile water and injected at the one-cell stage together with Tetramethylrhodamine Dextran (10,000 MW) or Fluoresceinated Dextran (FLDX) (70,000 MW) at 5 mg/ml. A dose-response curve was obtained and a concentration at which the oligomer did not elicit nonspecific defects was chosen. Approximately 2 to 4 pl of oligonucleotide solution were injected in the experiments described here.

*acvrII* Mo - `GGATCTTTCCCAGCCATTTCGGATA`

*bmp5/8* Mo - `CTTGGAGAGAAAATAAGCATATTCC`

The *bmp5/8* morpholino was used at 0.3 mM, while the *acvrII* morpholino was used at 0.5 mM. All the injections were repeated multiple times and for each experiment >50 embryos were analyzed (see S1 Table).

## In situ hybridization

The *nodal* and *chordin* probes have been previously described [5,6]. Probes derived from pBluescript vectors were synthesized with T7 RNA polymerase after linearization of the plasmids by NotI, while probes derived from pSport were synthesized with SP6 polymerase after linearization with SfiI. Control and experimental embryos were developed for the same time in the same experiments.

## Expression constructs and HEK293 cell transfections

For expression in HEK293 cells, the open reading frames of the desired constructs were cloned in the pCS2+ vector downstream of the CMV promoter. HEK cells were maintained in DMEM supplemented with 10% fetal bovine serum, penicillin, and streptomycin at respectively 100 µg/ml and 100 µg/ml at 37˚C and under 5% CO2. About 80,000 cells were seeded on a 110 mm tissue culture plate 18 h before transfection. Cells were transfected using the cationic polymer Polyethylenimine (PEI) as described by Boussif and colleagues [47]. PEI (800 kDa) (Fluka) was dissolved in water at 0.9 mg/ml stock in H2O, adjusted to pH 7 and filter sterilized. Immediately before transfection, the plasmids (4 or 8 µg) were diluted into 1 ml of DMEM (without FBS, Penicillin or Streptomycin) then a 3-fold amount of Polyethylenimine (12 or

24 μg) was added. After a 15-min incubation at room temperature, the transfection mixture was added to the plate of cells. A control plasmid coding for the Green Fluorescent Protein was used as control to estimate the efficiency of transfection. The medium was replaced 6 h post transfection to avoid toxicity. After 48 h, the cells were washed twice with phosphate buffer saline (PBS) and lysed with 1 ml of lysis buffer (50 mM Tris HCl (pH 7.5), 150 mM NaCl, 5% Glycerol, 1% Triton X100, 1 mM EDTA), supplemented with protease and phosphatase inhibitors cocktails from Roche (refs 04693124001 and 04906837001) for 20 min at 4°C. Lysates were centrifuged at 14,000g for 10 min at 4°C and their protein concentration was measured using the Bradford assay (Biorad). The supernatants were kept at −80°C.

## Co-immunoprecipitation

Cell lysates or culture supernatants were diluted to a working concentration of 0.5 μg/ml, and 2 μg of anti-Myc antibody was added to 100 μg of diluted lysate or supernatant and rotated overnight at 4°C. The next morning, magnetic beads were washed 3 times with lysis buffer (50 mM Tris HCl (pH 7.5), 150 mM NaCl, 5% Glycerol, 1% Triton X100, 1 mM EDTA), and then added to the lysates or supernatants. After rotation at 4°C for 2 h, the magnetic beads were washed 5 times with lysis buffer. Proteins retained by the beads were eluted by adding 2× Laemmli buffer + 100 mM DTT. The eluates were used for western blot after removal of the beads.

## Western blotting

Protein samples (10 μg/lane) were separated by SDS-gel electrophoresis and transferred to PVDF membranes (0.5 μm). After blocking in 5% dry milk blots were incubated overnight with the primary antibody diluted in 5% BSA in TBST. After washing and incubation with the secondary antibody, bound antibodies were revealed by ECL immunodetection using the SuperSignal West Pico Chemiluminescent substrate (Pierce) and imaged with a Fusion Fx7.

Antibodies used:

- Anti-Myc clone 9E10 (Thermo Fischer ref. 13–2500)

- Anti-Myc 71D10 (Cell Signaling, ref. 2278)

- anti-HA C29F4 (Cell Signaling, ref. 3724)

## Supporting information

**S1 Fig. (A) Overexpression of Panda does not suppress Smad1/5/8 signaling activated by overexpression of an activated BMP type I receptor Alk1/2 QD**. While overexpression of Panda suppresses BMP signaling activated by overexpression of *acvrII*, it does not suppress BMP signaling induced by misexpression of an activated BMP receptor. **(B) Overexpression of *acvrII* dorsalizes embryos in the absence of BMP5/8.** Figure shows wild-type embryo, embryo injected with a morpholino targeting the *bmp5/8* transcript, or embryo co-injected with the *bmp5/8* Morpholino and *acvrII* mRNA at the pluteus stage. Unlike *bmp5/8* morphants, which show a typical BMP loss-of-function phenotype with an ectopic ciliary band forming on the dorsal side, embryos injected with *bmp5/8* Mo and *acvrII* mRNA are dorsalized as indicated by their radialization, the presence of a thin ectoderm and the overpigmentation. This suggests that overexpression of *acvrII* dorsalizes embryos due to ligand independent activation of pSmad1/5/8 signaling.
(PDF)

**S2 Fig. Pro-domain deletion does not affect Panda function.** When mRNA encoding the mature form of Panda, *panda^mat^*, is injected locally into one blastomere at the two-cell stage, the progeny of the injected blastomere are on the dorsal side in 100% of the embryos injected indicating that removal of the pro-domain of Panda does not affect its activity.
(PDF)

**S3 Fig. Panda does not interact with Alk3/6.** Co-immunoprecipitation of Panda^mat^ with Alk3/6. Alk3/6 Myc co-immunoprecipitation in the presence of Panda^mat^ when probed with anti-HA fails to detect the presence of Panda^mat^, showing that Panda does not interact with Alk3/6.
(PDF)

**S4 Fig. Alignment of the mature ligand domain of Panda with those of other members of the TGF-β family.** Notice the presence of a Proline residue in the β6 region of both Panda and Lefty family members (red box) instead of a highly conserved Serine or threonine in Nodals, Activins, or BMPs (Blue box). Also notice that the region between the beta6 and beta7 strands is longer in Panda and Lefty family members compared to Nodal, Activin, and BMP ligands.
(PDF)

**S1 Table. Number of times the experiments were replicated.**
(XLSX)

**S1 Data. Raw data of the QPCR analysis.**
(XLSX)

**S2 Data. Full alignment of the TGF-β used for the phylogeny.** The full-length sequences of the 182 TGF-beta ligands were aligned using Clustal Omega with default parameters. The alignment was colorized using the Clustal color system.
(PNG)

**S1 Text. Abbreviations of the names used and accession numbers of TGF-β used for the phylogeny.**
(DOCX)

**S1 Phylogenetic Tree. Phylogenetic analysis shows that Panda belongs to a family of TGF-β that comprises Drosophila Maverick, Inhibin-alpha, and human GDF15.** Phylogenetic analysis of sea urchin and various metazoan TGF-beta ligands. The analysis was performed using the full-length proteins (see supplementary text). Maximum likelihood tree including 182 sequences. Reliability of for internal branches was assessed using the approximate likelihood ratio test (aLRT) test (SH-Like). Graphical representation and edition of the tree were performed with TreeDyn. The tree was calculated with the maximum likelihood method with PhyML with the substitution model WAG. A consensus tree with a 50% cut-off was derived using the aLRT test.
(PNG)

**S1 Raw Images. Raw images of the gels and blots.**
(PDF)

## Acknowledgments

We thank Julie Hanotel for help with biochemistry experiments and for her enthusiasm and continuous support of the project. We thank Thomas Lamonerie for access to the cell culture facility and for constant support. We thank Maximilian Fürthauer for interesting discussions about Panda.

## Author Contributions

**Conceptualization:** Thierry Lepage.

**Funding acquisition:** Thierry Lepage.

**Investigation:** Praveen Kumar Viswanathan, Aline Chessel, Maria Dolores Molina, Emmanuel Haillot, Thierry Lepage.

**Project administration:** Thierry Lepage.

**Supervision:** Thierry Lepage.

**Writing – original draft:** Praveen Kumar Viswanathan, Thierry Lepage.

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
