## [Editor Report · Decision Letter 0]

14 Sep 2023

Dear Dr Lepage, 

Thank you for submitting your manuscript entitled "The maternal TGF-β ligand Panda breaks the radial symmetry of the embryo by antagonizing the Nodal type II receptor ACVRII in Paracentrotus lividus" for consideration as a Research Article by PLOS Biology.

Your manuscript has now been evaluated by the PLOS Biology editorial staff as well as by an academic editor with relevant expertise and I am writing to let you know that we would like to send your submission out for external peer review. However, we would like to consider the manuscript as an Update Article, so please select that type of article from the dropdown menu when you submit your metadata (see below).

Before we can send your manuscript to reviewers, we need you to complete your submission by providing the metadata that is required for full assessment. To this end, please login to Editorial Manager where you will find the paper in the 'Submissions Needing Revisions' folder on your homepage. Please click 'Revise Submission' from the Action Links and complete all additional questions in the submission questionnaire.

Once your full submission is complete, your paper will undergo a series of checks in preparation for peer review. After your manuscript has passed the checks it will be sent out for review. To provide the metadata for your submission, please Login to Editorial Manager (https://www.editorialmanager.com/pbiology) within two working days, i.e. by Sep 18 2023 11:59PM.

Kind regards,

Ines

--

Ines Alvarez-Garcia, PhD

Senior Editor

PLOS Biology

---

## [Decision Letter · Decision Letter 1]

20 Oct 2023

Dear Dr Lepage,

Thank you for your patience while your manuscript entitled "The maternal TGF-β ligand Panda breaks the radial symmetry of the embryo by antagonizing the Nodal type II receptor ACVRII in Paracentrotus lividus" was peer-reviewed at PLOS Biology. It has now been evaluated by the PLOS Biology editors, an Academic Editor with relevant expertise, and by three independent reviewers. 

The reviews are attached below. As you will see, the reviewers find the conclusions interesting and adding significantly to the previous study, but they also raise several issues that would need to be addressed before we can consider the manuscript for publication. Reviewer 1 asks for a couple of experiments to take into consideration alternative explanations to the model and to confirm that Panda acts predominantly as a Nodal inhibitor in vivo. In addition, and in agreement with Reviewer 3, this reviewer would like you to confirm that Panda and Nodal directly interact. Reviewer 2 asks for several clarifications, but also thinks that the discussion is too speculative at this stage and should be streamlined. Reviewer 3 thinks it’ll be important to know if the biologically active form of Panda is a monomer or a dimer, and also confirming if Panda can directly activate ACVRII to confirm that it can act as an agonist (point 3).

In light of the reviews, we would like to invite you to revise the work to thoroughly address the reviewers' reports.

Given the extent of revision needed, we cannot make a decision about publication until we have seen the revised manuscript and your response to the reviewers' comments. Your revised manuscript is likely to be sent for further evaluation by all or a subset of the reviewers.

**IMPORTANT - SUBMITTING YOUR REVISION**

*Re-submission Checklist*

*Published Peer Review*

*PLOS Data Policy*

Please note that as a condition of publication PLOS' data policy (http://journals.plos.org/plosbiology/s/data-availability) requires that you make available all data used to draw the conclusions arrived at in your manuscript. If you have not already done so, you must include any data used in your manuscript either in appropriate repositories, within the body of the manuscript, or as supporting information (N.B. this includes any numerical values that were used to generate graphs, histograms etc.). For an example see here: http://www.plosbiology.org/article/info:doi%2F10.1371%2Fjournal.pbio.1001908#s5

*Blot and Gel Data Policy*

Sincerely,

Ines

--

Ines Alvarez-Garcia, PhD

Senior Editor

PLOS Biology

Reviewers' comments

Rev. 1:

Here, Viswanathan et al examine the mechanisms by which the atypical TGF-b ligand Panda antagonizes Nodal signaling within the early sea urchin embryo. Functional studies demonstrate that Panda blocks the activity of the TGF-b receptor ACVR2 when co-injected into the entire embryo or into a single blastomere. They further show that it interacts physically with ACVR2 as well as other TGF-b co-receptors including Cripto and the Inhibin receptor Betaglycan. Through phylogenetic comparisons with other TGF-b ligands and antagonists, they find that a single amino acid substitution is common to most antagonists, and that converting this residue to the conserved active form causes Panda to instead act as a Nodal agonist. Together, these data provide compelling evidence that Panda inhibits Nodal signaling through antagonism of ACVR2. However, additional questions remain, and addressing them with additional experimental evidence would strengthen the claims made in this manuscript. These points are enumerated below:

1. All the evidence presented is consistent with Panda acting as an ACVR2 antagonist, except one: Panda overexpression within 1-celled embryos does not phenocopy ACVR2 morphants and in fact has almost no phenotype at all. Why, if Panda blocks the activity of overexpressed ACVR2, does it not also block endogenous ACVR2 function? And how does Panda overexpression affect Nodal expression?

2. The authors show that ACVR2 activates ectopic pSmad1/5/8 even in the absence of BMP5/8 and conclude that, because no other BMPs are expressed at this time, that this activity of ACVR2 must be ligand independent. However, they also note that BMP2/4 is induced by Nodal at later stages of development. Isn't it possible then that ACVR2 overexpression accelerates induction of BMP2/4 by Nodal signaling and the increase in pSmad1/5/8 is dependent on this ligand? To help distinguish between these possibilities, the authors should examine expression of both BMP2/4 and Nodal upon ACVR2 overexpression.

3. Here, the authors find that the ability of Panda to promote dorsal fates upon injection into a single blastomere does not depend on BMP receptors. However, these findings are opposite to those reported by this same group in a previous study (Haillot et al, 2015). What explains this discrepancy?

4. The authors speculate that although ACVR2 can activate BMP upon overexpression and Panda can block this activity, Panda acts predominantly as a Nodal inhibitor in vivo. To confirm this experimentally, they could similarly test the ability of Panda to block pSmad1/5/8 activation by BMP ligands or Alk1/2 / Alk3/6 overexpression. If Panda cannot counteract BMP signaling through its canonical players, this would provide further support for this claim.

5. The authors further speculate that Panda may antagonize Nodal signaling by forming heterodimers with Nodal ligands and claim in the discussion to have preliminary data demonstrating a direct interaction between these ligands. These data should be included in this study.

6. Although embryo numbers are provided in the text and figures for some experiments, the number of embryos analyzed and independent experimental trials should be provided for all experiments, including co-IPs.

Minor point:

1. The authors state multiple times that ACVR2 is a limiting factor for Nodal signaling within the embryo. But if that were the case, why does its overexpression instead promote BMP signaling? To me, this suggests that Nodal is limiting and therefor, excess ACVR2 that is unoccupied by Nodal is then diverted to BMP.

Rev. 2:

This manuscript provides an update to the function of Panda, a TGF-ß family member, that is a maternally provided transcript in the sea urchin egg. Experiments here reinforce expand what is known about how Panda functions. Previously, based on indirect evidence, it was proposed that Panda was crucial for the establishment of the dorsal-ventral axis by setting up an early asymmetry in the function of Nodal. The proposal was that Panda acted through inhibition of two BMP receptors, Alk3/6 and Alk1/2, to restrict nodal expression to the ventral side, thereby establishing D-V symmetry. These experiments were indirect and did not fully establish that it was binding of Panda to the two BMP receptors, but was presented as a model of that possibility. Here, after further investigation, the Lepage group reports that the idea of Panda being involved in establishing D-V asymmetry by restricting Nodal expression to the ventral ectoderm is reinforced, but through a different receptor, ACVRII, a type II TGF-B receptor. Rather than indirectly proposing this as was done in the previous study, this group performed a number of biochemical studies to show that the binding of Panda was due to a direct interaction with ACVRII by way of a sequence that included a proline in the binding site, a finding that puts Panda in a class of ligands that includes Lefty, another Nodal antagonist. As such the updated information moves forward an important understanding of how the D-V axis in sea urchins is established to set up Nodal asymmetrical signaling. Further, although the data is not overwhelming, there is some evidence that maternal Panda is distributed asymmetrically in the egg, perhaps setting up the sequence of events that leads to the D-V asymmetry.

Below are editorial suggestions that this reviewer believes will improve the report. The one concern I have beyond these is the discussion. Much of the second half of the discussion is speculation on details of what might be going on in as yet unanswered questions. I recommend that this section be much reduced. The data in the Results section strongly supports the conclusions they reach. Yes, there are details remaining to be established, but speculation on what those details might yield are, in my opinion, unnecessary here. I suspect this group will do the experiments and it will be appropriate to learn from them when the speculation becomes information.

1. Ln 125-139. You cite two of your papers indicating that Nodal is expressed slightly before Lefty. That makes sense since Nodal is upstream of Lefty, but those two papers do not provide evidence to support the statement. Instead, they indicate, before you did a higher sensitivity in situ, that both genes are first expressed at the 60-cell stage.

2. Since most of your earlier papers referred to Alk4/5/7 as the Nodal receptor, it will be value on line 205-206 to clarify the relationship between that earlier descriptive and your current descriptive of ACVRII - and be consistent in the use of that or ACVR2. OK—I now come across your explanation of the relationship of all these molecules - so on line 205 you might add (see below) so the reader will put off his/her confusion - as I was confused at that point.

3. Line 257. You state that ACVRII overexpression was expected to activate Nodal signaling. Don't you mean you would expect it to augment Nodal signaling since Nodal is there?

4. On Figure 1C, ;The wild type 48 and 72h embryos are labeld lateral and vegetal, but they appear to be in the wrong position with the vegetal view in the lateral panel and the lateral view in the vegetal panel. Also, in Fig. 1 there should be a bar approximating the size of the embryo. It looks to me as though all the figures are about the same level of magnification so a single bar should suffice.

5. Line 310. The conclusion of this section is fine but it likely depends on a certain ratio of the two treatments. Is that the case? In other words, if you titre the Panda morpholino there is an expected series of outcomes with a titre much higher than the ACVRII overexpression eliminating all ACVRII activity. You seem to report the outcome that would be expected if Panda concentration erased the overexpression down to approximate the control level of expression.

6. Better stated on line 344: For this, acvrII mRNA along with a lineage tracer was injected into one blastomere of embryos at the two-cell stage.

7. Fig. 2b is confusing. The left panel shows the ventral side is the injected side. I suggest you put in outcome numbers on the figure. The middle panel shows what? And what are the numbers of that outcome? And the right panel is different from that but your figure legend and text don't indicate what it is. Are both outcomes of the ACVRII + panda injection? If so, include the numbers. But if so, how do you explain the far right image when you say Panda blocks the ability of ACVRII to impose a ventral identity? Fig. 2C provides the numbers but it would help if the 2B panel were better annotated perhaps by incorporating the perentages directly on the images with what I am assuming is Lateral andon the far right, ventral. Also, instead of saying the impact of ACVRII is inhibited, I would say it was cancelled.

8. Figure 2D and E are excellent. And also the table is excellent in supporting the hypothesized function of Panda.

Rev. 3:

The current studies provide clear biochemical and phenotypic evidence that the mechanism by which the maternally expressed TGFß family member Panda regulates nodal signaling is like that of Lefty, by sequestering ACVRII. They also show that Panda binds to Betaglycan, raising the possibility that it may use betaglycan as a binding partner to increase binding affinity for ACVRII, similar to Inhibins. The studies are rigorous and clearly presented for the most part but there are a few issues that should be addressed to improve the manuscript:

Major recommendations:

1. It's important to know if the biologically active form of Panda is a monomer or dimer, particularly in consideration of the possibility that PandaP461S may function as a dominant mutant (see point 4 below). The authors should repeat the experiments shown in Panel 3A under reducing and non-reducing conditions.

2. Does Panda co-IP with Type I nodal receptors? (ALK4,5,7)? This is more relevant than showing that it does not bind to

---

## [Decision Letter · Decision Letter 2]

15 May 2024

Dear Dr Lepage,

Thank you for your patience while we considered your revised manuscript entitled "The maternal TGF-β ligand Panda breaks the radial symmetry of the embryo by antagonizing the Nodal type II receptor ACVRII in Paracentrotus lividus" for publication as a Update Article at PLOS Biology. This revised version of your manuscript has been evaluated by the PLOS Biology editors, the Academic Editor and one of the original reviewers.

Based on the reviews and our Academic Editor's assessment, we are likely to accept this manuscript for publication, provided you satisfactorily address the data and other policy-related requests.

In addition, we would like you to consider a suggestion to improve the title:

"Maternal TGF-β ligand Panda breaks the radial symmetry of the sea urchin embryo by antagonizing the Nodal type II receptor ACVRII"

Please then add the sea urchin species to the abstract.

We expect to receive your revised manuscript within two weeks. 

*Published Peer Review History*

*Press*

Sincerely,

Ines

--

Ines Alvarez-Garcia, PhD

Senior Editor

PLOS Biology

DATA POLICY:

Fig. 6C

Many thanks for providing the uncropped gels for Fig. 4A-E. Please add also the raw gels for Fig. S3.

CODE POLICY

Per journal policy, if you have generated any custom code during the curse of this investigation, please make it available without restrictions upon publication. Please ensure that the code is sufficiently well documented and reusable, and that your Data Statement in the Editorial Manager submission system accurately describes where your code can be found. [IF APPLICABLE: As the code that you have generated to XXX is important to support the conclusions of your manuscript, its deposition is required for acceptance.]

Reviewers' comments

Rev. 3:

This manuscript is now acceptable for publication.

---

## [Editor Report · Decision Letter 3]

7 Jun 2024

Dear Dr Lepage,

Thank you for the submission of your revised Update Article entitled "Maternal TGF-β ligand Panda breaks the radial symmetry of the sea urchin embryo by antagonizing the Nodal type II receptor ACVRII" for publication in PLOS Biology. On behalf of my colleagues and the Academic Editor, Mary Mullins, I am delighted to let you know that we can in principle accept your manuscript for publication, provided you address any remaining formatting and reporting issues. These will be detailed in an email you should receive within 2-3 business days from our colleagues in the journal operations team; no action is required from you until then. Please note that we will not be able to formally accept your manuscript and schedule it for publication until you have completed any requested changes.

PRESS

Sincerely, 

Ines

--

Ines Alvarez-Garcia, PhD

Senior Editor

PLOS Biology
